

# SurEau-Ecos v2.0: A trait-based plant hydraulics model for simulations of plant water status and drought-induced mortality at the ecosystem level

Julien Ruffault[1], François Pimont[1], Hervé Cochard[2], Jean-Luc Dupuy[1] and Nicolas Martin-StPaul[1]

[1]INRAE, URFM, 84000 Avignon, France
[2]Université Clermont Auvergne, INRAE, PIAF, 63000 Clermont-Ferrand, France

*Correspondence to*: Nicolas Martin-StPaul (nicolas.martin@inrae.fr)

**Abstract.** A widespread increase in tree mortality has been observed around the globe, and this trend is likely to continue because of ongoing climate-induced increases in drought frequency and intensity. This raises the need to identify regions and

ecosystems that are likely to experience the most frequent and significant damages. We present *SurEau-Ecos*, a trait-based, plant-hydraulic model designed to predict tree desiccation and mortality at scales from stand to region. *SurEau-Ecos* draws on the general principles of the process-based, soil-plant-atmosphere *SurEau* model. It also introduces a simplified representation of plant architecture and alternative numerical schemes; both additions made to facilitate model parameterization and large-scale applications. In *SurEau-Ecos*, the water fluxes from the soil to the atmosphere are represented through two plant organs

(a leaf and a stem, which includes the volume of the trunk, roots and branches) as the product of an interface conductance and the difference of water potentials. Each organ is described by its symplasmic and apoplasmic compartments. The dynamics of plant's water status beyond the point of stomatal closure are explicitly represented via residual transpiration flow, plant cavitation and solicitation of plants' water reservoirs. In addition to the "explicit" numerical scheme of *SurEau*, we implemented a "semi-implicit" and "implicit" scheme. Both schemes led to a substantial gain in computing time compared to

the "explicit" scheme (>10,000 times), and the implicit scheme was the most accurate. We also observed similar plant water dynamics between *SurEau-Ecos* and *SurEau* but slight disparities in infra-daily variations of plant water potentials that we attributed to the differences in the representation of plant architecture between models. A global model's sensitivity analysis revealed that factors controlling plant desiccation rates differ depending on whether leaf water potential is below or above the point of stomatal closure. Total available water for the plant, leaf area index, and the leaf water potential at 50% stomatal

closure mostly drove the time needed to reach stomatal closure. Once stomata are closed, resistance to cavitation, residual cuticular transpiration and plant water stocks mostly determined the time to hydraulic failure. Finally, we illustrated the potential of *SurEau-Ecos* to simulate regional drought-induced mortality over France. *SurEau-Ecos* is a promising tool to perform regional-scale predictions of drought-induced hydraulic failure, determine the most vulnerable areas and ecosystems to drying conditions, and asses the dynamics of forest flammability.



# 1. Introduction

Forests across many regions worldwide are experiencing record-breaking droughts followed by widespread increase in climate-driven disturbance events, including tree mortality (Allen et al., 2015; Fettig et al., 2019; Schuldt et al., 2020), wildfires (Abram et al., 2021; Ruffault et al., 2020) and insect outbreaks (Jactel et al., 2012). Droughts are likely to become more frequent and more intense over the next decades because of the global increase in temperatures and heatwaves coupled, in some regions, to some changes in the hydrological cycle (Trenberth et al., 2014). Given the importance of forests for biochemical cycles and ecosystem services (Seidl et al., 2014), there is a growing need for the development of models that can simulate the response of forests to extreme drought. Process-based vegetation models can help to address these issues because they represent the mechanisms governing plant physiological responses to drought and account for the interspecific and intraspecific variations of tree traits and their potential acclimation to a rapidly changing climate.

The science of plant hydraulics seeks to understand the physical and physiological mechanisms driving water transport in plants. This research field has proven to be a relevant theoretical framework to study the effect of global changes on plant and the terrestrial water cycle (Brodribb et al., 2020; Choat et al., 2018). Advances in plant hydraulic modelling have accelerated over the last two decades (Fatichi et al., 2016; Mencuccini et al., 2019) and used as mean to tackle diverse prediction challenges, such as tree mortality (De Kauwe et al., 2020; Venturas et al., 2020), water use efficiency (De Cáceres et al., 2021; Domec et al., 2017) or species distribution (Sterck et al., 2011). Many of SPA models were also designed (or reformatted) to be integrated into land-surface models and improve the representation of the feedbacks between land and climate systems (Christoffersen et al., 2016; Kennedy et al., 2019; Li et al., 2021; Xu et al., 2016). Recently, modeling water transport in plants also proved to be a promising way to assess the seasonal dynamics of live fuel moisture (foliage and twigs water content, dead to live fuel ratio), a key variable for fire behavior that could play a major role in raising forests' flammability under climate warming (Nolan et al., 2020; Ruffault et al., 2018).

Most plant hydraulic models represent water fluxes in plants through a mathematical approach of soil-plant-atmosphere continuum, wherein diffusion laws control the water flow through the soil, root, and leaves (Mencuccini et al., 2019). Water flow through plants is considered as being analogous to the electric current through a circuit with a series of resistance and/or capacitance (Sperry et al., 1998). SPA models, however, vary widely in their complexity, some of them representing trees as a single resistance (Mackay et al., 2003; Williams et al., 1996), while others include multiple resistances and capacitances (Couvreur et al., 2018; Sperry et al., 1998; Tuzet et al., 2017). How physiological processes regulate plant transpiration also differs between SPA models (Mencuccini et al., 2019). Some models describe stomatal conductance through semi-empirical models (Christoffersen et al., 2016; Feng et al., 2018; Li et al., 2021; Williams et al., 1996) while others are based on optimality approaches (Sperry et al., 2017; Wang et al., 2020).

The *SurEau* SPA model was developed specifically to simulate plant desiccation under extreme drought and heatwaves (Cochard et al., 2021; Martin-StPaul et al., 2017). As other SPA models, *SurEau* describes the soil-plants-atmosphere system as a network of resistances and capacitances and computes water exchanges until stomatal closure.



Additionally, *SurEau* simulates plant tissue desiccation beyond the point of stomatal closure by accounting for residual plant transpiration and the discharge of internal plant water stores (Fig. 1a). Unlike most current approaches (e.g., Tuzet et al., 2017;
Xu et al., 2016), *SurEau* explicitly accounts for the differences in capacitance of the symplasmic and apoplasmic compartments, which can be calibrated, from pressure-volume curves for the symplasm and vulnerability curves for the apoplasm. Symplasmic capacitances mostly buffer water fluxes during well-watered conditions, whereas apoplasm capacitances come into play when cavitation occurs (Fig. 1a). Thus, *SurEau* accounts for the leading role of cavitation in the dynamics of plant desiccation (Mantova et al., 2021) and the probability of plant mortality (Adams et al., 2017). *SurEau* has
been successfully evaluated against field cavitation observations (Cochard et al. 2021; hereafter CPRM21), applied in different contexts (Lemaire et al., 2021; López et al., 2021), and performed well in predicting plant water fluxes when compared to other plant hydraulic models (Mcdowell et al., in press).

As noted in CPRM21, two *SurEau*'s characteristics impede its use for large-scale ecological applications or its integration into terrestrial biosphere models. First, SurEau requires a high number of parameters because of its detailed representation of plant
architecture, and of the mechanisms involved in plant water exchanges. The second limitation of *SurEau* is its high computation time, which is partly due to the use of a first-order "explicit" numerical scheme to compute water flows. This scheme requires that variations in water quantities be computed at very small-time steps to avoid numerical instabilities due to the Courant-Friedrichs-Lewy condition (CFL; Dutykh, 2016). A numerical method has been proposed to overcome these instabilities and increases the time step (Tuzet et al., 2017; Xu et al., 2016) but this is not directly compatible with *SurEau*'s specificities
regarding capacitances and cavitation. Moreover, knowledge regarding numerical physics and methods for simulation have seldom been applied to plant hydraulics.

We present *SurEau-Ecos,* a new SPA model meant to improve the predictions of ecosystems' transpiration, desiccation and drought-induced mortality at scales from stand to region. *SurEau-Ecos* draws on the physiological and physical framework of *SurEau* while limiting the number of parameters and reducing computational cost. In the following sections, we first describe
the principles, functioning, main equations and numerical schemes of *SurEau-Ecos*. Second, we compare simulations produced with three numerical schemes (Explicit, Semi-implicit and Implicit) in terms of predictions' stability and computing time. Third, we further describe the differences in plant hydraulic architecture between *SurEau-Ecos* and *SurEau* (CPRM21) and their impacts on simulation results. Fourth, we perform a global sensitivity analysis of tree desiccation dynamics to the main *SurEau-Ecos* input, i.e., plant hydraulic traits and stand and soil parameters. Fifth, we illustrate the potentialities which *SurEau-*
*Ecos* will provide by running prospective simulations of hydraulic failure probability at the regional scale under changing climate.

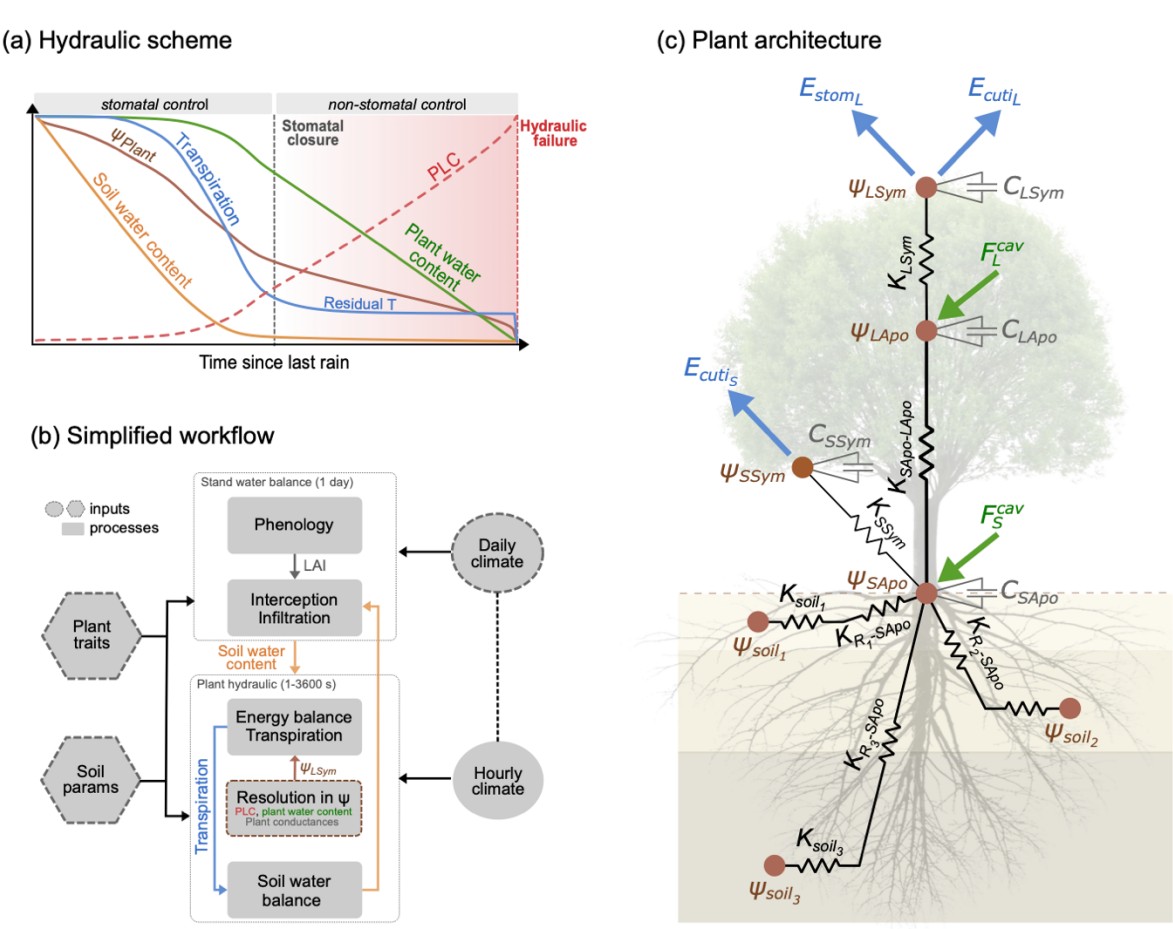

**Fig. 1: Overview of the *SurEau-Ecos* plant-hydraulic model. (a) Schematic trajectories of the main processes involved in drought-induced tree mortality under extreme drought. In a first phase, stomata are open and transpiration gradually empty soil water reservoirs. Then stomata gradually close as water potential decreases. In a second phase, once stomata are fully closed, only residual transpiration (equivalent to cuticular transpiration in the model) remains. Percent loss of conductivity (PLC) increases and the plant mostly rely on internal water reservoirs until hydraulic failure. (b) Simplified workflow of *SurEau-Ecos*. Key modules and their interactions are shown by arrows and boxes. (c) Schematic representation of the plant hydraulic architecture in *SurEau-Ecos*.**

## 2. Description of SurEau-Ecos

### 2.1 Model overview

*SurEau-Ecos* is a plant hydraulic model that simulates water fluxes between the soil, plant and atmosphere for a monospecific layer of vegetation. In *SurEau-Ecos* the soil-plant system is discretized into three soil layers and two plants compartments: a leaf and a "stem" (Fig. 1c). Each of the two plant organs contains an apoplasm and a symplasm. The stem apoplasm and symplasm include water volumes of all non-leaf compartments, i.e., trunk, root and branches.





Water dynamics of the soil-plant-atmosphere system (represented by nodes in Fig. 1c) are locally governed by a generic partial
differential equation for water mass conservation:

$$\frac{dq}{dt} = \nabla.(k\nabla\psi) + s \tag{1}$$

where $q$ is the local water content, $k$ the conductivity, $\psi$ the water potential, $k\nabla\psi$ the water fluxes and $s$ the local sink term
(i.e., a negative sign for soil evaporation or transpiration) or source term (i.e., a positive sign for precipitation and water released
by cavitation).

A spatially-integrated form of Eq. (1) can be specified for each compartment of the plant (Fig. 1c) to derive the rate of change
of its total water content. To account for the water fluxes between compartments and for the contribution of internal water
stocks (i.e., capacitances). The computations of water fluxes between two adjacent compartments ($F_{i \rightarrow j}$) are simulated
according to Darcy's law as the product of compartment' interface conductance ($K_{ij}$) and the gradient of water potential ($\psi$) :

$$F_{i \rightarrow j} = k_{ij}\nabla\psi \approx K_{ij}(\psi_j - \psi_i) \tag{2}$$

These fluxes are described in Sect. 2.3.

In addition, solving Eq. 1 needs to describe the link between Q and $\psi$. This is handled using the notion of capacitance for the
plant compartments and water retention curves for the soil compartments. Plant capacitances ($C$) are defined as:

$$C = \frac{dQ}{d\psi} \tag{3}$$

For any plant compartments a generic equation of the water balance can now be written:

$$C\frac{d\psi_i}{dt} + \sum_j K_{ij}(\psi_i - \psi_j) - S = 0 \tag{4}$$

According to the type of compartment, $S$ includes cuticular or stomatal transpiration losses or water release from cavitation
which is also accounted as a source term in the apoplasm (Cruiziat et al., 2002). Cuticular or stomatal transpiration fluxes are
computed differently for each compartment (leaf symplasm includes stomatal transpiration whereas stem symplasm only
include cuticular transpiration). The contribution of capacitance ($C$) to the plant compartment water balance is related to the
saturated (or initial) water quantity ($Q^{sat}$) in that compartment and takes different formulation for symplasm and apoplasm. A
pressure-volume curve is used for the symplasmic capacitance (Tyree and Hammel, 1972) whereas a constant capacitance is
used for the apoplasm (Sec. 2.5). To the best of our knowledge, this is the first formulation of symplasmic $C$ and cavitation
flux as Darcy's law (see details in Sec. 2.3.3. and 2.5.1). These generic forms are needed for the numerical resolution of water
balance at each plant node (described in Sect. 2.2.1).

For soil compartments, the water balance of a soil layer $j$ is computed using a generic equation following Eq. 1 and 2 such as:





$$\frac{dQ_{soil_j}}{dt} + K_{soil_j-SApo}\left(\psi_{soil_j} - \psi_{SApo}\right) - S = 0 \qquad (5)$$

Where $K_{soil_j-Sapo}$ is the conductance from the soil layer $j$ to the stem apoplasm (Sect. 2.3.1). $S$ represents a source (when S>0)

or sink (when S<0) term that can include soil water inputs from soil infiltration, drainage from other layer or outputs such as deep drainage, soil evaporation or capillarity depending on the soil layer (Sect. 2.2.2). A water retention curve for the soil (van Genuchten, 1980) is used to link $Q_{soil_j}$ and $\psi_{soil_j}$ and solve eq. 5 (Sect. 2.5.2).

In addition to the core soil-plant hydraulic processes driving transpiration and plant water status ($Q$ and $\psi$), SurEau-Ecos also includes an empirical module for leaf phenology that control leaf area growth and decrease during senescence (described in

Appendix A) and different modules to represent the stand water balance (interception, water transfers between soil layers and drainage; described in Ruffault et al. 2013). The list of input variables and their respective unit is given in Tab. 1.

Temporal resolution varies according to each type of process (Fig. 1b). Phenology and stands water balance are computed at a daily time step. Soil-plant hydraulics processes (i.e., soil water uptake, transpiration and hydraulic redistribution) are computed at the finer time step (from 0.01 to 1800 s depending on the resolution scheme) and driven by hourly interpolated

climate, which is derived from daily climate following De Cáceres et al., (2021) (See Tab. B1 for the list of daily input weather variables). The three different numerical resolution schemes currently implemented in *SurEau-Ecos* are described in Sect. 2.6. All variables and processes related to stand water balance processes (precipitation, interception, drainage) are expressed per unit ground surface area, while plant hydraulic processes are expressed per unit leaf surface area, in accordance with usual practices in each research field. This implicates that initial water volumes of the soil and the plant (leaf and stem) are expressed

per unit soil area. Then, leaf area index (LAI) permits to convert quantities from a soil area basis to a leaf area basis. If the parametrization is performed from individual tree dimensions or from forest inventories and allometries, an additional parameter is needed, the average plant foot print (*aPFP*, in m$^2$), in order to scale individual plant dimensions on leaf or a soil area basis.

*SurEau-Ecos* was implemented in the *R* programming language (R Core Team, 2020). The following sections describe the

equations and resolution of the model in more details.



| | Parameter | Description | Unit |
|---|---|---|---|
| **Stand** | $LAI_{max}$ | Maximum leaf area index of the stand | $m^2_{leaf}.m^{-2}_{soil}$ |
| | $t_0$ | Initial date of the forcing period for leaf phenology | $DOY$ |
| | $T_D$ | Minimum temperature to start cumulating temperature for budburst | $°C$ |
| | $F^*$ | Amount of forcing temperature to reach budburst | $°C$ |
| | $R_{LAI}$ | LAI growth rate per day | $LAI.day^{-1}$ |
| | $cws$ | Canopy water storage capacity | $mm.LAI^{-1}$ |
| | $k$ | Light extinction parameter | - |
| **Plant** | $\varepsilon_L$ | Modulus of elasticity of the leaf symplasm | $MPa$ |
| | $\pi_{0_L}$ | Osmotic potential at full turgor of the leaf symplasm | $MPa$ |
| | $\varepsilon_S$ | Modulus of elasticity of the stem symplasm | $MPa$ |
| | $\pi_{0_S}$ | Osmotic potential at full turgor of the stem symplasm | $MPa$ |
| | $slope_L$ | Slope of rate of leaf embolism spread at $\psi_{50,L}$ | $\%.MPa^{-1}$ |
| | $\psi_{50,L}$ | Water potential causing 50% loss of leaf hydraulic conductance | $MPa$ |
| | $slope_S$ | Slope of rate of stem embolism spread at $\psi_{50,S}$ | $\%.MPa^{-1}$ |
| | $\psi_{50,S}$ | Water potential causing 50% loss of stem hydraulic conductance | $MPa$ |
| | $K_{R-SApo,max}$ | Maximum conductance from the root surface to the stem apoplast | $mmol.m^{-2}_{leaf}.s^{-1}.MPa^{-1}$ |
| | $K_{SApo-LApo,max}$ | Maximum conductance from trunk apoplasm to the leaf apoplasm | $mmol.m^{-2}_{leaf}.s^{-1}.MPa^{-1}$ |
| | $K_{SSym}$ | Conductance from the stem apoplasm to stem symplasm | $mmol.m^{-2}_{leaf}.s^{-1}.MPa^{-1}$ |
| | $K_{LSym}$ | Conductance from the leaf apoplasm to leaf symplasm | $mmol.m^{-2}_{leaf}.s^{-1}.MPa^{-1}$ |
| | $\alpha_{LApo}$ | Leaf apoplasmic fraction | - |
| | $\alpha_{SApo}$ | Stem apoplasmic fraction of the wood water volume | - |
| | $\alpha_{SSym}$ | Stem symplasmic fraction of the wood water volume | - |
| | $C_{LApo}$ | Capacitance of the leaf apoplasm | $mmol.m^{-2}_{leaf}.MPa^{-1}$ |
| | $C_{SApo}$ | Capacitance of the stem apoplasm | $mmol.m^{-2}_{leaf}.MPa^{-1}$ |
| | $V_S$ | Volume of tissue of the stem (includes the root, trunk and branches) | $L.m^{-2}_{soil}$ |
| | $Succulence$ | Leaf succulence (water content per unit leaf area) | $g.m^{-2}_{leaf}$ |
| | $LDMC$ | Leaf dry mater content (dry mass over saturated mass) | $g.g^{-1}$ |
| | $LMA$ | Leaf mass per area | $g.m^{-2}_{leaf}$ |
| | $\beta$ | Shape parameter for root distribution | - |
| | $RaLa$ | Root to leaf area ratio | - |
| | $d_R$ | Root diameter | $m$ |
| | $\psi_{gs50}$ | Water potential causing 50% stomatal closure | $MPa$ |
| | $slope_{gs}$ | Rate of decrease in stomatal conductance at $\psi_{gs,50}$ | $\%.MPa^{-1}$ |
| | $g_{stom\_min}$ | Minimum stomatal conductance | $mmol.m^{-2}_{leaf}.s^{-1}$ |
| | $g_{stom\_max}$ | Maximum stomatal conductance | $mmol.m^{-2}_{leaf}.s^{-1}$ |
| | $\delta$ | Response of $g_{stom}$ to light | - |
| | $T_{optim}$ | Temperature at maximal stomatal conductance | $°C$ |
| | $T_{sens}$ | Stomatal sensitivity to temperature | $°C$ |
| | $g_{crown0}$ | Reference crown conductance | $mmol.m^{-2}_{leaf}.s^{-1}$ |
| | $g_{cuti20}$ | Cuticular conductance at 20°C | $mmol.m^{-2}_{leaf}.s^{-1}$ |
| | $Q_{10a}$ | Temperature dependance of $g_{cuti}$ when $T \leq T_{Phase}$ | - |
| | $Q_{10b}$ | Temperature dependance of $g_{cuti}$ when $T > T_{Phase}$ | - |
| | $T_{Phase}$ | Temperature for transition phase of $g_{cuti}$ | $°C$ |
| **Soil** | $rfc_j$ | Rock fragment content of layer $j$ | $\%$ |
| | $d_j$ | Maximum depth of layer $j$ | $m$ |





| $\theta_s$ | Soil water content at saturation | - |
|---|---|---|
| $\theta_r$ | Residual soil water content | - |
| $\alpha$ | Inverse of the air entry potential | $MPa^{-1}$ |
| $n$ | Pore size distribution index | - |
| $l$ | Shape parameter for the Van Genuchten equation | - |
| $k_{sat}$ | Soil hydraulic conductivity at saturation | $mmol.m_{soil}^{-1}.s^{-1}.MPa^{-1}$ |
| $g_{soil0}$ | Reference soil conductance to water vapor | $mmol.m_{soil}^{-2}.s^{-1}$ |

**Tab. 1: Input parameters in *SurEau-Ecos.***

## 2.2. Water balance in each compartment

### 155 2.2.1. Plant

The water balance of each of the four-plant compartment (leaf and stem symplasm and apoplasm, Fig. 1c) is determined according to the generic Eq. 4 and solved at each time step.

For the leaf apoplasm, the water balance equation is:

$$\underbrace{C_{LApo}\frac{d\psi_{LApo}}{dt}}_{Water\ quantity\ change} + \underbrace{K_{SApo-LApo}\left(\psi_{LApo}-\psi_{SApo}\right)}_{Flux\ to\ stem\ apoplasm} + \underbrace{K_{LSym}\left(\psi_{LApo}-\psi_{LSym}\right)}_{Flux\ to\ leaf\ symplasm} - \underbrace{F_L^{cav}}_{Cavitation} = 0 \qquad (6)$$

The first term represents the change in water quantity related to the leaf apoplasmic capacitance ($C_{LApo}, mmol.m_{leaf}^{-2}.MPa^{-1}$)

which releases or absorbs water according to volume changes due to water potential changes ($\psi_{LApo}$, MPa). Contrary to symplasmic compartments, this term is very limited in the apoplasm because by xylem wall is inelastic. Note also that cavitation is not included in this capacitance. The second and third terms are the water exchanges between the leaf apoplasm and stem apoplasm and between the leaf apoplasm and leaf symplasm, respectively. $\psi_{SApo}$ is the water potential of the stem apoplasm, $\psi_{LSym}$ is the water potential of the leaf symplasm, $K_{SApo-LApo}$ ($mmol.m_{leaf}^{-2}.s^{-1}.MPa^{-1}$) is the conductance from

the stem apoplasm to leaf apoplasm and $K_{LSym}$ is the conductance of the leaf symplasm. This equation applies to the non-cavitated part of the xylem, which receives water from the cavitated part. This source is represented by the fourth term $F_L^{cav}$ (*mmol*), which corresponds to the water release by the cavitated vessels towards the non-cavitated leaf apoplasm (Hölttä et al., 2009). This term is further described in Sect. 2.3.2, where we explain how it can be expressed as a function of $\psi_{LApo}$.

Water balance for the stem apoplasm:

$$\underbrace{C_{SApo}\frac{d\psi_{SApo}}{dt}}_{Water\ quantity\ change} + \underbrace{\sum_j K_{soil_j-SApo}\left(\psi_{SApo}-\psi_{soil_j}\right)}_{Flux\ to\ soil\ layers} + \underbrace{K_{SApo-LApo}\left(\psi_{SApo}-\psi_{LApo}\right)}_{Flux\ to\ leaf\ apoplasm} + \underbrace{K_{SSym}\left(\psi_{SApo}-\psi_{SSym}\right)}_{Flux\ to\ stem\ symplasm} - \underbrace{F_S^{cav}}_{Cavitation} = 0 \qquad (7)$$

The first term represents the water flux related to the stem apoplasmic capacitance ($C_{SApo}$) and water potential ($\psi_{SApo}$) changes during the time step. As with the leaf apoplasm, this term is in general very limited for the stem apoplasm. The second term represents the water exchange between the stem apoplasm and the three soil layers. For each soil layer $j$, $K_{soil_j-SApo}$ is the conductance from the soil to the stem apoplasm and $\psi_{s_j}$ the soil water potential. The third and fourth terms represent flux to





the leaf apoplasm and stem symplasm, respectively. $\psi_{SSym}$ is the water potential of the stem symplasm and $K_{SSym}$ is the stem-symplasm conductance. The fifth term $F_S^{cav}$, corresponds to the water released from cavitation to the non-cavitated stem apoplasm water reservoir.

Water balance for the leaf symplasm:

$$\underbrace{C_{LSym}\frac{d\psi_{LSym}}{dt}}_{Water\ quantity\ change} + \underbrace{K_{LSym}(\psi_{LSym} - \psi_{LApo})}_{Flux\ to\ leaf\ apoplasm} + \underbrace{E_{stom}}_{Stom\ transpiration} + \underbrace{E_{cuti_L}}_{Leaf\ cuticular\ transpiration} = 0 \qquad (8)$$

The first term represents the water flux related to $C_{LSym}$ and water potential changes of the leaf symplasm ($\psi_{LSym}$) during the time step. The second term is the exchange between leaf apoplasm leaf symplasm. The third and fourth terms represent the losses of water from the plant to the atmosphere, through leaf stomatal transpiration ($E_{stom}$) and cuticular leaf transpiration ($E_{cuti_L}$). Note that with this formulation, leaf water losses from leaf transpiration remains lower-bounded by $E_{cuti_L}$ even when stomata are fully closed ($E_{stom} = 0$)

Water balance for the stem symplasm:

$$\underbrace{C_{SSym}\frac{d\psi_{Ssym}}{dt}}_{Water\ quantity\ change} + \underbrace{K_{SSym}(\psi_{SSym} - \psi_{SApo})}_{Flux\ to\ stem\ apoplasm} + \underbrace{E_{cuti_S}}_{Stem\ cuticular\ transpiration} = 0 \qquad (9)$$

The first term represents the water flux related to $C_{SSym}$ and water potential changes of the stem symplasm ($\psi_{SSym}$) during the time step. The second term is the flux to the stem apoplasm. The third and fourth terms represent the losses of water from the plant to the atmosphere, through minimum cortical transpiration ($E_{cuti_S}$).

### 2.2.2 Soil

The water balance of each of the three soil layers (Fig. 1c) is determined according to the generic equation 5 and solved at each time step.

For the first soil layer:

$$\frac{dQ_{soil_1}}{dt} + \underbrace{k_{soil_1-SApo}(\psi_{soil_1} - \psi_{SApo}).LAI}_{Flux\ to\ Stem\ Apoplasm} + ppt_{soil} - D_{1\rightarrow2} - E_{soil} = 0 \qquad (10)$$

The first term ($\frac{dQ_{soil_1}}{dt}, mmol.m_{soil}^{-2}$) represents the change in soil water quantity between two consecutive time steps. The second term is the flux to the stem apoplasm. This flux is multiplied by LAI to convert water quantities from a leaf area basis to a soil area basis. $ppt_{soil}$ ($mmol.m_{soil}^{-2}$) is the precipitation reaching the soil. $D_{1\rightarrow2}$ the drainage ($mmol.m_{soil}^{-2}$) of the first to the second layer, and $E_{soil}(mmol.m_{soil}^{-2})$ is soil evaporation that occurs only from this layer.

Similarly, for the second layer:

$$\frac{dQ_{soil_2}}{dt} + \underbrace{K_{soil_2-SApo}(\psi_{soil_2} - \psi_{SApo}).LAI}_{Flux\ to\ Stem\ Apoplasm} + D_{1\rightarrow2} - D_{2\rightarrow3} = 0 \qquad (11)$$

For the third soil layer:





$$\frac{dQ_{soil_3}}{dt} + \underbrace{K_{Soil_3-SApo}\big(\psi_{soil_3} - \psi_{SApo}\big).LAI}_{Flux\ to\ Stem\ Apoplasm} + D_{2\to3} - Dd = 0 \tag{12}$$

Where $Dd$ is the deep drainage ($mmol.m^{-2}_{soil}$). For any layer, drainage occurs when the field capacity of the soil layer ($\theta_{fc}$) is overpassed. Lateral water transfer processes and upward capillary transfers between layers are neglected. At the time step of the hydraulic model ($dt$) the water balance of each soil layer is treated according to the losses from transpiration and from evaporation (only for the first layer). Incoming fluxes from precipitation, drainage and transfers between soil layers are treated at a daily time step (Fig. 1b). Rainfall interception and drainage are treated as in SIERRA (Mouillot et al., 2001; Ruffault et al., 2013), and follows the design principles of several other water balance models (De Cáceres et al., 2015; Granier et al., 1999; Rambal, 1993).

## 2.3 Conductances and fluxes

### 2.3.1 Plant and soil conductances

The model includes four apoplasmic conductances (three root-to-stem and one stem-to-leaf), two symplasmic conductances (one for the stem and one for the leaves) and three soil conductances (one per soil layer) (Fig. 1c). Symplasmic conductances of the leaves ($K_{LSym}$) and stem ($K_{SSym}$) drive the fluxes between the symplasmic and apoplasmic compartments. These conductances are set at a constant value throughout the simulation. Xylem (i.e., apoplasmic) conductances are composed of three root-to-stem conductances in parallel (one per soil layer) and of a trunk-to-stem conductance. These conductances can vary throughout the simulation from their initial value down to 0 according to the level of cavitation (expressed by the percent loss in conductance).

The stem-to-leaf apoplasmic conductance ($K_{SApo-LApo}$) is expressed as a function of the percent loss of conductance due to xylem embolism in the leaf:

$$K_{SApo-LApo} = K_{SApo-LApo,max}\frac{100 - PLC_L}{100} \tag{13}$$

Where $k_{SApo-LApo,max}$ is the initial (maximum) root to leaf conductance and $PLC_L(\%)$ is the percent loss of conductance. $PLC_L$ is proportional to the level of xylem embolism. It occurs when the water potential drops below the capacity of the leaf xylem to support negative water potential and is computed by using the sigmoidal function (Pammenter and Vander Willigen, 1998) :

$$PLC_L = \frac{100}{1 + e^{\left(\frac{slope_L}{25}.(\psi_{LApo} - P_{50,L})\right)}} \tag{14}$$

Where $P_{50,L}$ (MPa) is the water potential causing 50% loss of plant hydraulic conductance and $slope_L$ (%/MPa) is the slope of linear rate of embolism spread per unit water potential drop at the inflexion point $P_{50,L}$.

The apoplasmic conductance from each root $j$ to the stem apoplasm ($K_{R_j-SApo}$) is expressed as a function of the level of embolism computed at the node of the stem apoplasm:





$$K_{R_j-SApo} = K_{R_j-SApo,max} \frac{100 - PLC_{SApo}}{100} \qquad (15)$$

Where $PLC_S$ is computed as $PLC_L$ with the stem apoplasmic potential ($\psi_{SApo}$) and vulnerability curves parameters specific to the stem ($slope_S$ and $P_{50,S}$). $K_{R_j-SApo,max}$ is the maximal root-to-stem apoplasmic conductance of layer $j$. It is derived from

225 fine root area of the layer $j$ such as:

$$K_{R_j-SApo,max} = RAI_j \times K_{R-SApo} \qquad (16)$$

Where $K_{R-SApo}$ is the total conductance of the root system. $RAI_j$ is the fine root area of the layer $j$:

$$RAI_j = RAI \times r_j \qquad (17)$$

Where $RAI$ is the total fine root area which is computed from the stand leaf area index and the root to leaf area ratio $RaLa$; and $r_i$ the root fraction in each soil layer which is determined according to the equation from Jackson $et\ al.$ (1996):

$$r_i = \left( \beta^{z_{h,j-1} \cdot 100} - \beta^{z_{h,j} \cdot 100} \right) \qquad (18)$$

Where $z_{h,j}$ is the depth ($m$) from the soil surface to the interface between layers $j$ and $j + 1$, the factor 100 converts from m to

230 cm, and $\beta$ is a species-dependent root distribution parameter (Jackson et al. 1996). Then, the conductance between each soil layer $j$ and the stem apoplasm ($K_{soil_j-Sapo}$) is determined as the result of two conductances in series, $K_{R_jT}$ and the conductance from soil to root ($K_{soil_j-R_j}$):

$$K_{soil_j-SApo} = \frac{1}{\frac{1}{K_{R_j-SApo}} + \frac{1}{K_{soil_j-R_j}}} \qquad (19)$$

The conductance of the soil to fine roots $K_{soil_j-R_j}$ for each soil layer $j$ is computed as:

$$K_{soil_j-Rj} = \frac{2\pi L_{a,j}}{ln\left(\frac{1}{r\sqrt{\pi L_{v,j}}}\right)} k_{sat} REW_j \left[ 1 - (1 - REW_j^{\frac{1}{m}})^m \right]^2 \qquad (20)$$

with $L_a$ and $L_v$ the root length per soil area and soil volume for each soil layer respectively, and are computed from soil depth

and $RAI_j$ whereas $r$ is the radius of fine absorbing roots. $k_{sat}$ is the soil hydraulic conductivity at saturation, $m$ is a parameter of shape from the Van-Genuchten equation and $REW$ is the relative extractable water content computed as:

$$REW = \frac{\theta - \theta_r}{\theta_s - \theta_r} \qquad (21)$$

With $\theta$ the relative water content (soil water content per unit soil volume) changing dynamically with changes in absolute soil water reserve in the rooting zone, $\theta_s$ is the relative soil water content at saturation and $\theta_r$ is the relative soil water content at wilting point. $\theta_s$ and $\theta_r$ are parameters measured in the laboratory or derived from soil surveys with pedotransfer functions.





### 2.3.2 Cavitation

*SurEau-Ecos* also considers the capacitive effect of cavitation (Hölttä et al., 2009), i.e., the water released to the streamflow when cavitation occurs. The non-cavitated part of the xylem receives a water flux from the cavitated part, corresponding to $F_L^{cav}$ in Eq. 6 ($F_L^{cav} > 0$), and then transferred to adjacent compartments. The amount of water corresponding to a new cavitation event is derived from the quantity of water in the apoplasm at saturation ($Q_{LApo}^{Sat}$) and the temporal variations in $PLC_L$ such as:

$$F_L^{cav} = Q_{LApo}^{Sat} \max\left(\frac{dPLC_L}{dt}, 0\right) \tag{22}$$

This flux is linearized in temporal variations in $\psi_{LApo}$ in order to express this flux in the form of a Darcy's law to match the generic form of equation 2. For that purpose, we introduce an equivalent conductance ($K_L^{cav}$) such as:

$$F_L^{cav} = Q_{LApo}^{Sat} \frac{dPLC}{d\psi}\frac{d\psi}{dt} \approx K_L^{cav} max\left(0, \psi_{LApo}^{cav} - \psi_{LApo}\right) \tag{23}$$

With $K_L^{cav} = -\frac{Q_{LApo}^{Sat} PLC'(\psi_{LApo})}{dt}$, $PLC'$ the derivative of the PLC with respect to $\psi$ which is computed from cavitation curve and $\psi_{LApo}^{cav}$ the minimal value of potential ever reached over time, which controls the current cavitation level ($PLC_L = PLC_L(\psi_{LApo}^{cav})$). $PLC'$ is computed as:

$$PLC' = -\frac{slope}{25}\frac{PLC}{100}\left(1 - \frac{PLC}{100}\right) \tag{24}$$

Following the same approach, the flux derived from the stem when cavitation occurs is defined as:

$$F_S^{cav} = Q_{SApo}^{Sat} \frac{dPLC_S}{dt} \approx K_T^{cav} max\left(0, \psi_{SApo}^{cav} - \psi_{SApo}\right) \tag{25}$$

### 2.4 Sources and Sink

#### 2.4.1 Stomatal and cuticular plant transpiration

Plant loses water through stomatal transpiration ($E_{stom}$), cuticular transpiration of the leaf ($E_{cuti_S}$) and cuticular transpiration of the stem ($E_{cuti_S}$). Cuticular transpiration of the roots is considered to be negligible and is not taken into account. The total plant transpiration $E_{Plant}$ is decomposed as the sum of the leaf ($E_{leaf}$) and wood transpiration ($E_{cuti_S}$):

$$E_{Plant} = E_L + E_{cuti_S} \tag{26}$$

Where $E_{leaf}$ is computed as:

$$E_L = E_{stom} + E_{cuti_L} = \frac{1}{\frac{1}{g_{stom} + g_{cuti_L}} + \frac{1}{g_{bound}} + \frac{1}{g_{crown}}} \cdot \frac{VPD_{leaf}}{P_{atm}} \tag{27}$$

And $E_{cuti_S}$ is computed as:





$$E_{cuti_S} = \frac{1}{\frac{1}{g_{cuti_S}} + \frac{1}{g_{bound}} + \frac{1}{g_{crown}}} \frac{VPD_S}{P_{atm}} \qquad (28)$$

With $VPD_L (MPa)$ the vapor pressure deficit of the leaf, $P_{atm}$ the atmospheric pressure (MPa), $g_{stom}$ the stomatal conductance, $g_{cuti_L}$ the cuticular conductance of the leaf, $g_{bound}$ the conductance of the leaf boundary layer and $g_{crown}$ the conductance of the tree crown.

$VPD_L$ is a function of leaf temperature ($T_L$). $T_L$ is computed at the leaf surface by solving the energy budget as in CPRM21. $g_{bound}$ and $g_{crown}$ are computed following Jones, (2013). $g_{bound}$ varies with leaf shape, size ($d_{leaf}$) and wind speed;
$g_{crown}$ is a function of wind speed.

$g_{cuti_L}$ is a function of $T_L$ which is based on a single or double $Q_{10}$ equation depending on whether leaf temperature ($T_L$) is above or below the transition phase temperature ($T_{Phase}$) (Cochard, 2019):

$$if\ T_L \leq T_{phase}\quad g_{cuti_L} = g_{\text{cuti20}_L} Q_{10a}^{\frac{T_L - 20}{10}} \qquad (29)$$

$$if\ T_L > T_{phase}\quad g_{cuti_L} = g_{\text{cuti20}_L} Q_{10a}^{\frac{T_{phase} - 20}{10}} Q_{10b}^{\frac{T_L - T_{phase}}{10}} \qquad (30)$$

Where $g_{stom}$ is the stomatal conductance taking into account the dependence of $g_{stom}$ to light, temperature, and $CO_2$
concentration on the one hand, and water status on the other such as:

$$g_{stom} = \gamma \cdot g_{stom,max} \qquad (31)$$

$g_{stom,max}$ is the stomatal conductance without water stress and is determined as a function of light, temperature and $CO_2$ concentration following Jarvis (1976). $\gamma$ is a regulation factor that varies between 0 and 1 to represent stomatal closure according to $\psi_{LSym}$ and an empirical sigmoid function depending on the potential at 50 % of stomatal closure ($\psi_{gs,50}$) and a shape parameter ($slope_{gs}$) describing the rate of decrease in stomatal conductance per unit water potential drop.

$$\gamma = 1 - \frac{1}{1 + e^{\frac{slope}{25}(\psi_{LSym} - \psi_{gs50})}} \qquad (32)$$

**2.4.1 Soil evaporation**

$E_{soil}$ is computed as the minimum of two supply function that both depend on maximum soil conductance ($g_{soil0}$) and the REW of the first soil layer such as

$$E_{soil} = min(g_{soil0} \cdot REW_1 \cdot \frac{VDP}{P_{Atm}}, g_{soil0} \cdot REW_1 \cdot (PET \cdot e^{-k.LAI})) \qquad (33)$$

Where the fraction of radiative energy (Potential evapotranspiration, PET) reaching the soil depends on the light extinction coefficient ($k$).





## 2.5. Capacitances

As described in Sect. 2.1, the link between Q and $\psi$ are not represented in the same way for the soil and plant compartments. The notion of capacitance is used for the plant while water retention curves are used for the soil.

### 2.5.1. Plant compartments

The contribution of capacitance ($C$) to the plant compartment water balance is related to the saturated (or initial) water quantity ($Q$) in that compartment. Symplasmic and apoplasmic capacitances are not modelled in the same way but both require the water volume at saturation ($Q^{sat}$) of the considered reservoir. For the leaves, the volume of symplasmic and apoplasmic reservoirs at saturation ($Q_{LSym}^{Sat}$ and $Q_{LApo}^{Sat}$, respectively) are defined as:

$$Q_{LSym}^{Sat} = (1 - \alpha_{LApo})Q_L^{Sat} \tag{34}$$

$$Q_{LApo}^{Sat} = \alpha_{LApo}Q_L^{Sat} \tag{35}$$

$$with\ Q_L^{Sat} = \frac{1}{LDMC - 1}DM = succulence \tag{36}$$

where DM is the dry matter per unit leaf area and the leaf dry matter content (LDMC), fraction of apoplasmic tissue in the leave ($\alpha_{LApo}$) and leaf mass per area (LMA) are all input parameters.

The apoplasmic and symplasmic water quantities of the stem at saturation ($Q_{SSym}^{Sat}$ and $Q_{SApo}^{Sat}$, respectively) includes the volume includes the roots, trunk and branches. They are computed based on the volume of the woody compartment and the water fraction of this volume such as:

$$Q_{SSym}^{Sat} = \frac{V_S}{M_{H_2O}} . \alpha_{Water} . \alpha_{SSym} \tag{37}$$

$$Q_{SApo}^{Sat} = \frac{V_S}{M_{H_2O}} . \alpha_{Water} . \alpha_{SApo} \tag{38}$$

Where $V_S$ is the volume of tissue of the stem compartment (including the root, trunk and branches), $M_{H_2O}$ is the water molar mass, $\alpha_{Water}$ is the proportion of water in this volume and $\alpha_{SApo}$ and $\alpha_{SSym}$ are the apoplasmic and symplasmic fraction of this water volume, respectively.

Symplasmic reservoirs behave as variable plant capacitances related to the pressure volume curve, which corresponds to the water quantity changes in symplasmic cells ($\frac{dQ}{dt}$). Symplasmic conductances are functions of the $Q_{LSym}^{Sat}$ and the temporal change in the symplasmic relative water content ($RWC$) (illustrated here for the leaf but similar equations apply for the trunk):

$$\frac{dQ}{dt} = Q_{LSym}^{Sat}\frac{dRWC}{dt} = Q_{LSym}^{Sat}\frac{dRWC}{d\psi_{LSym}}\frac{d\psi_{LSym}}{dt} = C_{LSym}\frac{d\psi_{LSym}}{dt} \tag{39}$$

With this formulation the capacitance of the leaf symplasm ($C_{Lsym}$) can be written as:





$$C_{Lsym} = Q_{LSym}^{Sat} RWC'$$ (40)

Where $RWC'$ the derivative of the $RWC$ with respect to $\psi_{LSym}$, derived from pressure-volume curves (Bartlett et al., 2012; Tyree and Hammel, 1972):

$$\psi_{LSym} = \begin{cases} -\pi_0 - \epsilon(1 - RWC) + \dfrac{\pi_0}{RWC}, & RWC \geq \dfrac{\pi_0}{\epsilon} + 1 \\ \dfrac{\pi_0}{RWC}, & RWC \leq \dfrac{\pi_0}{\epsilon} + 1 \end{cases}$$ (41)

We used the following formulation for $RWC'$ (see justification below for the expression above $\psi_{tlp}$):

$$RWC' = \begin{cases} \dfrac{RWC}{-\pi_0 - \psi_{LSym} - \epsilon + 2\epsilon RWC}, & \psi_{LSym} \geq \psi_{tlp} = \dfrac{1}{\dfrac{1}{\epsilon} + \dfrac{1}{\pi_0}} \\ -\dfrac{\pi_0}{\psi_{LSym}^2}, & \psi_{LSym} < \psi_{tlp} \end{cases}$$ (42)

With

$$RWC = \frac{\pi_0 + \epsilon + \psi_{LSym} + \sqrt{(\pi_0 + \epsilon + \psi_{LSym})^2 - 4\epsilon\pi_0}}{2\epsilon}$$ (43)

When $\psi_{Lsym} \geq \psi_{tlp} = \dfrac{1}{\frac{1}{\epsilon} + \frac{1}{\pi_0}}$

In the above equation the formulation for $\psi_{LSym} < \psi_{tlp}$ simply results from the fact that $RWC = \dfrac{\pi_0}{\psi_{LSym}}$.

The case $\psi_{LSym} \geq \psi_{tlp}$ was obtained from basic manipulations of the derivation of the following form of the pressure volume curve:

$$\psi_{LApo} RWC + \pi_0 RWC + \epsilon(1 - RWC)RWC - \pi_0 = 0$$ (44)

Which derivative with respect to $\psi_{LSym}$ is:

$$\psi_{LSym} RWC' + RWC + \pi_0 RWC' + \epsilon(1 - RWC)RWC' - \epsilon RWC RWC' = 0$$ (45)

So that $RWC' = \dfrac{RWC}{-\pi_0 - \psi_{LApo} - \epsilon + 2\epsilon RWC}$

Apoplasmic capacitance are constant and are computed as the product between $Q_{Apo}^{Sat}$ and the specific apoplasmic capacitance ($C_{Apo}$). Note that given the very low elasticity of the xylem, their contribution is very weak.

**2.5.2 Soil compartments**

Capacitances for soil are not explicitly computed in *SurEau-Ecos*. Rather, soil water potentials for the different soil layers ($\psi_{soil}$, $MPa$) are directly computed according to the Van Genuchten parametric formulation (van Genuchten, 1980):



$$\psi_{soil} = \frac{\left(\left(\frac{1}{REW}\right)^{\frac{1}{m}} - 1\right)^{\frac{1}{n}}}{\alpha} \tag{46}$$

where $m$, $n$ and $\alpha$ are empirical parameters describing the typical sigmoidal shape of the function and REW is the relative extractable water (see equation 21)

## 2.6. Numerical resolution

**2.6.1. Plant compartments**

The resolution of the plant hydraulic part of *SurEau-Ecos* is to solve the water balance for the four hydraulic compartments (i.e., nodes in Fig. 1c) whose equation are presented in Sect. 2.2.1. Three different numerical resolution schemes were implemented to solve water balances of plant compartments. For these three schemes, water potentials were discretized between two consecutive time steps $\psi^n$ and $\psi^{n+1}$, separated by $dt$. Thanks to cautious hypotheses, these equations were

linearized at the first order in $\psi$, to lead to a four-equations linear system. In particular, we neglected all variations of capacitances and conductances during a given time step ($C \approx C^n$ and $K \approx K^n$), as these variations are expected to be marginal, with respect to weather changes, stomatal regulation or water release by cavitation.

The simpler *"explicit scheme"*, also implemented in *SurEau*, assumes that water fluxes can be expressed from the current time step $n$ (see appendix B1 for details). From the generic water balance Eq. 4, it leads to:

$$C\frac{\psi^{n+1} - \psi^n}{dt} + \sum_j K_j\left(\psi^n - \psi_j^n\right) + S^n = 0 \tag{47}$$

Rearranging this equation, the potential at the next time step $\psi^{n+1}$ can be simply computed as:

$$\psi^{n+1} = \psi^n + \frac{dt}{C}\left(\sum_j K_j\left(\psi_j^n - \psi^n\right) - S^n\right) \tag{48}$$

While the implementation of the explicit time integration scheme is undoubtedly the most straightforward numerical solution, it suffers, however, from a well-known numerical constraint referred to as the CFL, which imposes very small-time steps (dt) to avoid numerical instabilities:

$$dt \le \frac{C}{2\,max(K_j)} \tag{49}$$

This constraint implies that the smaller the $C$, the smaller the $dt$. An intuitive interpretation of this limitation is that the time

step needs to be small enough to avoid water movements between not adjacent cells. This constraint is particularly strong in plant xylem that is inelastic (i.e., C is very small) such that apoplasmic compartments cannot absorb water fluxes from their adjacent compartments when the time step is too large. This typically imposes time steps smaller than 10 ms (CPRM21).



A common option to avoid these numerical instabilities is to use an *"implicit scheme"*, where fluxes are estimated from the values of $\psi$ at time $n+1$ ($\psi^{n+1}$) such as:

$$C\frac{\psi^{n+1} - \psi^n}{dt} + \sum_j K_j(\psi^{n+1} - \psi_j^{n+1}) + S^n = 0 \tag{50}$$

This numerical scheme is unconditionally stable, meaning that an increase in $dt$ will not induce numerical instabilities but might however induce a loss of numerical accuracy. One very important limitation of this scheme is that the equations of the different compartments now correspond to a system of four equations that are coupled. Such a system can be linearized (by pieces to account for thresholds such as cavitation) and solved. In general, it implies the inversion of the matrix of the linear system, but the resolution can also be done analytically when the equations are not too many, as it is the case with *SurEau-*

*Ecos* (see details Appendix B2).

An alternative scheme, based on a "*semi- implicit"* approach, has also been recently proposed to solve water balances in plant hydraulic models while overcoming the numerical instabilities associated with an explicit formulation (De Kauwe et al., 2020; Li et al., 2021; Tuzet et al., 2017; Xu et al., 2016). Although not usual in numerical resolution approaches, this scheme showed great performance and led to convergence in simulations with time steps on the order of 10 minutes (Xu et al., 2016).

This approach consists in solving the differential equation of each compartment assuming that $\psi_j$ and $S$ remain constant (respectively equals to $\psi_j^n$ and $S^n$) such as:

$$C\frac{d\psi}{dt} + \sum_j K_j(\psi - \psi_j^n) + S^n = 0 \tag{51}$$

After linearization of the coefficient, this ordinary differential equation has the following solution:

$$\psi(u) = \psi^n e^{-\frac{\sum_j K}{C}u} + \left(1 - e^{-\frac{\sum_j K_j}{C}u}\right)\frac{\sum_j K_j \psi_j^n - S^n}{\sum_j K_j} \tag{52}$$

So that $\psi^{n+1}$ can be estimated by its value at $u = dt$:

$$\psi^{n+1} = \psi(dt) \tag{53}$$

Which implies that $\psi^{n+1} = \eta\psi^n + (1 - \eta)\tilde{\psi} \tag{54}$

With $\eta = e^{-\frac{\sum_j K_j}{C}dt} \tag{55}$

And $\tilde{\psi} = \frac{\sum_j K_j \psi_j^n - S^n}{\sum_j K_j} \tag{56}$

One can notice here that $\tilde{\psi}$ is the steady state solution of the equation, typically valid when C=0 (Fully elastic media).

In practice, this formulation is equivalent to the corresponding numerical scheme (provided that $dt$ is very small):

$$C\frac{\psi^{n+1} - \psi^n}{dt} + \sum_j K_j(\psi^{n+1} - \psi_j^n) + S^n = 0 \tag{57}$$

This formulation allows comparing this scheme to the explicit and implicit schemes proposed above. This scheme uses $\psi^{n+1}$ as a value for $\psi$ (so that it remains stable) and $\psi_j^n$ as a value of $\psi_j$ (so that the equations of the four compartments are





decoupled) and can be seen as an intermediate between the explicit and the implicit scheme. For that reason, it will be referred to as *"semi-implicit"* (Appendix B3). In theory, the water fluxes computed from values of water potentials evaluated at
different time steps should be less accurate than the implicit scheme, especially when water potential changes are fast. It is thus expected that simulations require a larger time step to converge than the implicit scheme.

For the three different numerical schemes, we assume that soil potentials were estimated at the current time step $n$ (i.e., $\psi_{S_j} \approx \psi_{S_j}^n$ ) as in the explicit formulation (instead of $n+1$, as normally expected in an implicit scheme). This assumption is supported by the very small variations in soil potentials occurring during a single time and avoids the linearization of soil potential
equations which would have required unnecessary complex developments.

Source and sink fluxes $S^{n+\frac{1}{2}}$ are computed for the climate at the middle of the time step (mid climate between current and next time step, at $n + \frac{1}{2}$). For the implicit scheme, to account for the quick adjustment of stomatal regulation to climate variations, $E^{n+\frac{1}{2}}$ accounts linear variations in water potential $\psi_{LSym}$ over the time step, thanks to the derivative of transpiration function $S'^{n+\frac{1}{2}}(\psi_{LSym}^n)$, also estimated for the mid climate, but current regulation $\psi_{LSym}^n$:

$$S^{n+\frac{1}{2}}\left(\psi_{LSym}^{n+\frac{1}{2}}\right) \approx S^{n+\frac{1}{2}}\left(\psi_{LSym}^n\right) + S'^{n+\frac{1}{2}}\left(\psi_{LSym}^n\right)\left(\frac{\psi_{LSym}^{n+1} - \psi_{LSym}^n}{2}\right)$$

$$= S^{n+\frac{1}{2}} + \frac{S'^{n+\frac{1}{2}}}{2}\left(\psi_{LSym}^{n+1} - \psi_{LSym}^n\right)$$

(58)

### 2.6.2. Soil compartments

Soil water balance in *SurEau-Ecos* is solved for each soil layer (Sect. 2.2.2) following a simple explicit scheme assuming that water fluxes can be expressed from the current time step $n$. From the generic soil water balance Eq. 5, it leads to:

$$\frac{Q^{n+1} - Q^n}{dt} + K_{soil_j - SApo}\left(\psi_j^n - \psi_{SApo}^n\right) + S^n = 0$$

(59)

### 3. Impacts of numerical schemes on simulations and computation times

In this section, we explore the benefits and limitations of the three numerical schemes implemented in *SurEau-Ecos* to solve
water fluxes, namely an "explicit", "semi-implicit' and "implicit" scheme. As mentioned above, the minimal time step required for accurate simulations is determined by computational limitations that depend on the chosen scheme. First, unlike the implicit and semi-implicit scheme, the "explicit" scheme is limited by the CFL which causes numerical instabilities. We explored how much computation time can be gained by using implicit or semi-implicit schemes compared to the explicit. In addition, in the case of the implicit and semi-implicit scheme, reducing the temporal resolution (i.e., increasing the time step) can also limit
the accuracy of the simulation. The magnitude of corresponding errors then depends on the physiological processes at play in



the plant and on the precision of the numerical scheme. We also assessed the sensitivity of model outputs to the temporal resolution (time step $dt$) for the implicit and semi-implicit schemes.

For these simulations, all inputs were set identical to those used in the section dedicated to the evaluation of *SurEau-Ecos* (see Sect. 4). Daily weather was kept constant, without precipitation, and simulations were run until total hydraulic failure of the plant. To compare the explicit scheme with the two other schemes, we made two slight simplifications to the model. First, we neglected the cavitation term in Eq.s 6 and 7. Indeed, the explicit numerical scheme of *SurEau-Ecos* cannot account for the flux term associated with water released by cavitation. This is due to the direct dependence of $K_L^{cav}$ and $K_S^{cav}$ to $dt$ (Sect. 2.3.2) that prevents the CLF to be satisfied at any time step. Second, the values stem and leaf of apoplasmic capacitances ($C_{SApo}$ and $C_{LApo}$ ) were increased (from about 1e⁻³ to 10 $mmol.m^2.MPa$) to decrease computational costs and ease the comparison between the numerical schemes. The CFL constraint imposed very small-time steps (on the order of 1e-5 s) with the original values of plant apoplasmic capacitance, which caused unaffordable computation times under most CPUs. Preliminary analyses showed that the impact of $C_{SApo}$ and $C_{LApo}$ were negligible on simulation results for values up to 50-100 $mmol.m^2.MPa$.

When using the implicit or semi-implicit schemes with a relatively small-time step ($dt$ =10s) our results show that these schemes yielded identical plant dynamics to those obtained with the explicit mode (Fig. 2). However, the gains in computation time were considerable. Computation time was divided by about 10 for the implicit and semi-implicit scheme compared to the explicit scheme. This is because $dt$ had to be set to 1s for the explicit theme because of the CFL. Any attempt to set a $dt$ above this 1s threshold caused (as expected by the CFL) critical numerical instabilities (Fig. B1). Since some modifications to the model had to be performed for this comparison, the differences in computation times were solely indicative, reported to illustrate the benefits of the semi-implicit and implicit schemes compared to the explicit scheme. Our results showed that the semi-implicit scheme was less accurate than the implicit scheme. Smaller time steps were required for the convergence of the model. Numerical explorations show that the semi-implicit scheme requires time steps on the order of 1 min (which is slightly slower than described in Xu *et al.* (2016) which stated that 10 minutes was enough), whereas the time step can be generally larger than 30 minutes with the implicit scheme (Figs. B2 and B3).

| Resolution scheme | Time step | Computational time (s) |
|---|---|---|
| Explicit | 1s | 1403.98* |
| Implicit /Semi-implicit | 10s | 138.29 |
| | 1min | 21.42 |
| | Adaptative 'normal' (10-1min) | 4.78 |
| | 10 min | 3.35 |
| | Adaptative 'fast' (60-10 min) | 1.49 |

**Tab. 2: Comparison of computation times between the three resolution schemes implemented in *SurEau-Ecos*. *This computational time is for indicative purposes only as several changes had to be made to the model to run it with the "explicit" scheme (see details in main text).**

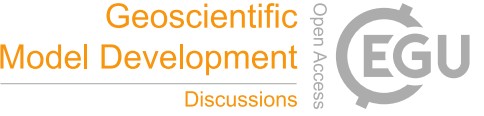

For the implicit and semi-implicit schemes, two adaptive time steps were further implemented to reduce computation times. This improvement was based upon the assumption that smaller time steps were only required when changes in two critical processes, stomatal regulation and cavitation, were the highest. In a *"Normal"* mode, the base time step is at 10 minutes but is automatically and gradually refined up to 1 min in periods of intense regulation changes, based on a criterion aiming at preventing variation in stomatal regulation and cavitation of more than 1% between two consecutive time steps. In a *"Fast"*

mode, the base time step is at 1h refined up to 10 minutes. The implementation of adaptative time steps allowed to further increase this gain in computing time (Tab. 2) without affecting plant dynamics.

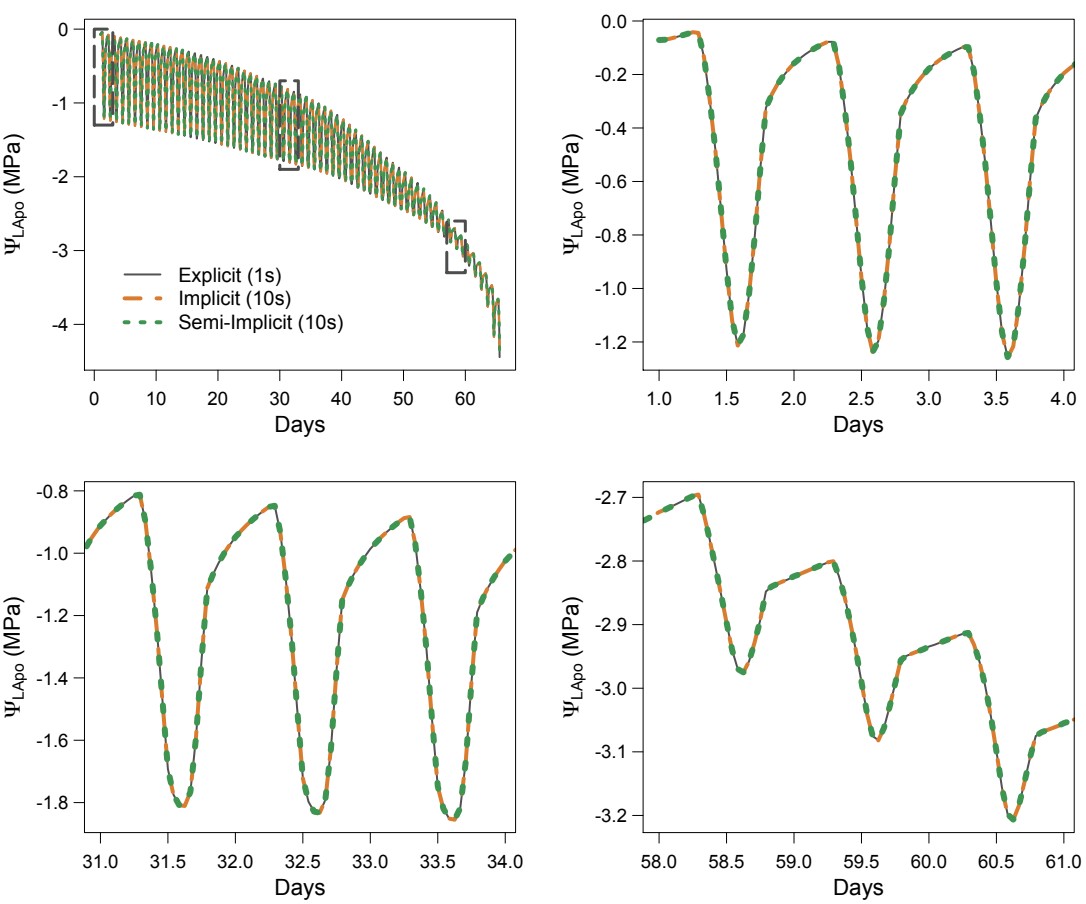

**Fig. 2: Comparison of the three numerical schemes implemented in *SurEau-Ecos* to solve water balances. Computation times for each scheme are given in Tab. 2.**




## 4. Comparison between *SurEau-Ecos* and *SurEau*

*SurEau-Ecos* relies on the same biological and physical principles of *SurEau* (CPRM21). The soil-plant-atmosphere system is segmented and described as compartments linked together and exchanging water fluxes according to the gradients of water potential and hydraulic conductances. However, significant disparities between the implementation, parametrization, and

resolution of water fluxes between the two models lead to some major differences in plant architecture and representation of water fluxes. It was therefore essential to confirm that both models provide comparable dynamics of the main state variables under similar conditions. This comparison of model outputs also consists at as an indirect validation effort of *SurEau-Ecos* since *SurEau* has been validated against field data (see details in CPRM21).

We identified three major differences in plant architecture and representation of hydraulic processes within the models. First

plant architecture is simpler in *SurEau-Ecos* than in SurEau. *SurEau-Ecos* represents the plant as two leaf cells (leaf apoplasm and leaf symplasm) and two stem compartments that include the woody volume of branches, trunk and root. By contrast, *SurEau* offers a detailed plant organ discretization (including roots, trunk, branches, leaves and buds). Second, while both models represent the belowground stems by three roots in parallel, the resistance to water flow linked to the root endoderm (a symplasmic root resistance) is not explicitly included in *SurEau-Ecos* contrary to *SurEau*. Instead, only one resistance per

root, from the root entry to the stem is accounted to mimic all possible resistance (root symplasm and apoplasm). Finally, in *SurEau-Ecos,* all leaf level fluxes to the atmosphere -- i.e., the stomatal and the cuticular fluxes -- pass through the symplasm, whereas in *SurEau* stomatal fluxes pass through the apoplasm and cuticular fluxes.

To compare model outputs, we performed an equivalent parameterization of the two models (see details in Fig. B4) and ran simulations until total hydraulic failure of the plant. We started the comparison with a typical plant fully described in CPRM21

whose parameters are given for each organ in Tab. B2. We then aggregated the values of *SurEau* parameters to match the following input parameters of *SurEau-Ecos*: water quantities of the leaf and stem compartments ($Q_{LApo}^{sat}$, $Q_{LSym}^{sat}$, $Q_{SApo}^{sat}$ and $Q_{SSym}^{sat}$), the symplasmic conductance of the stem ($K_{SSym}$), the apoplasmic root to stem conductance ($K_{R-SApo}$) and the apoplasmic stem to leaf conductance ($K_{SApo-LApo}$). We also set the cuticular conductance of non-leaf organ to 0 in both models. All other sub models, parameters and environmental forcing (weather and soil) were also set equal including stomatal,

boundary layer and crown conductance, linear approximation for the leaf energy balance, soil parameters and hourly climatic inputs. This ensured that any divergence between models could only come from either the numerical scheme or plant hydraulic architecture.

Fig. 3 shows the dynamics of water potentials, leaf transpiration and percent loss of conductance obtained when simulations were run from a wet soil profile until hydraulic failure is reached. Note that for this comparison the output of the trunk in

*SurEau* were compared to the stem in *SurEau-Ecos*. For both models, at the beginning of the simulations, when the soil was wet, leaf and stem water potentials followed the hourly variations in meteorological conditions, thereby reflecting the response of stomata to light and response of plant transpiration to $g_{stom}$ and VPD. As soil reservoir emptied, stomata progressively closed according to the intensity of foliar water potential. After about 65 days for both models, the stomata permanently closed



and transpiration was limited to cuticular losses which gradually accentuated the drought stress of the plant (decreased plant
water potentials). Simultaneously, cavitation increased in the different organs, inducing water release from the apoplasm which
partly dampened the decrease in plant water potentials. These results show that *SurEau-Ecos* and *SurEau* yielded very similar
results when parametrized in such a way that plant organs had by similar conductances and water reservoirs.

Despite similar dynamics, we also identified some differences in infra daily water potentials between the two models. As a
result, the time to leaf hydraulic failure was underestimated by three days (out of 90 days) in *SurEau-Ecos* compared to *SurEau*.
These slight differences can be linked to the presence of the higher number of compartments in *SurEau* that increase the
seasonal dampening effect of water potential compared to SurEau-Ecos where in a lower number of compartments are
represented. Notably, we observed some differences between the short-term (infra daily) variations of water potential dynamic
of the trunk symplasmic compartment of *SurEau* and the stem compartment of *SurEau-Ecos* (that includes the volume of roots,
trunk and branches, Tab. B2). The daily magnitude of the fluctuation in *SurEau-Ecos* appeared more dampened (Fig. 3, Figs
B5 and B6). The most plausible explanation for this difference is that the volume of the stem compartment in *SurEau-Ecos* is
greater than the volume of the trunk compartment in *SurEau*. This is likely to lead to greater water discharge and lower water
potential fluctuations in *SurEau-Ecos* (Fig. B6). Ongoing developments of a modular version of *SurEau* within the *Capsis*
modeling platform (Dufour-Kowalski et al., 2012) will allow to evaluate more deeply the effects of plant hydraulic architecture
on the dynamics of plant desiccation.

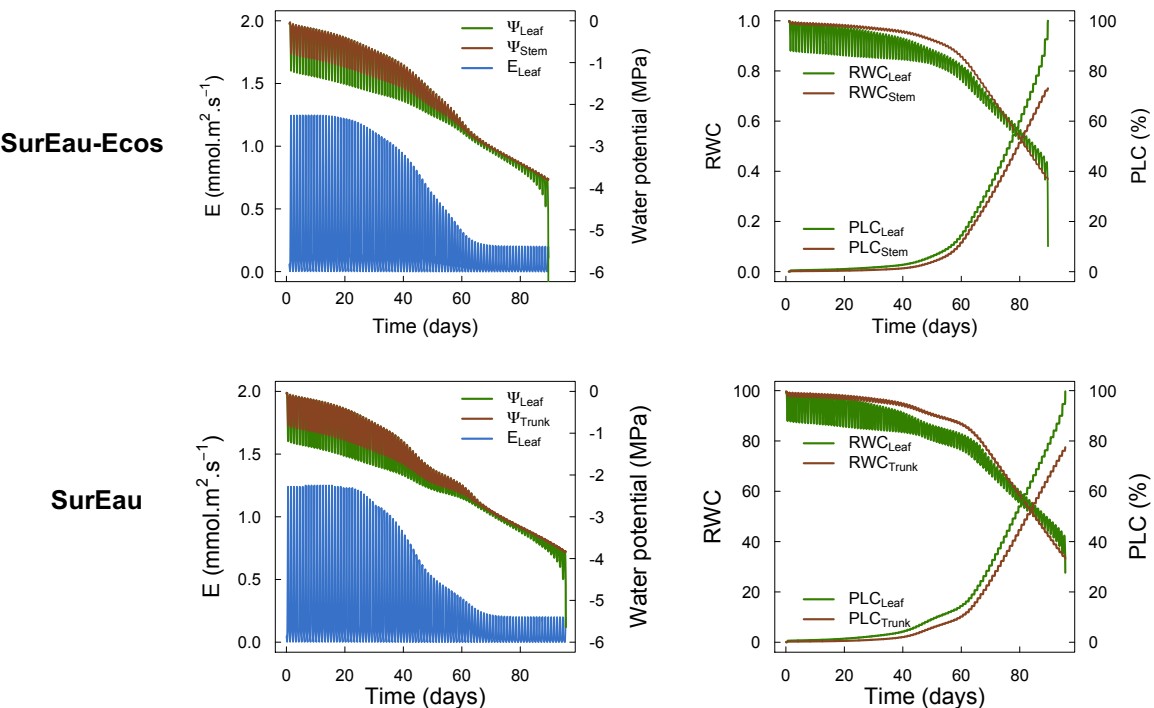


**Fig. 3. Comparison of the dynamics of plant water status between *SurEau-Ecos* and *SurEau*.**





## 5. Sensitivity analyses of drought-induced tree mortality to input parameters

We carried out a variance-based sensitivity analysis to gain insight into the hydraulic traits and stand parameters that influence plant dynamics in *SurEau-Ecos* and to explore the main drivers of tree resilience to extreme drought. Variance-based
approaches can measure sensitivity across the whole input space (i.e., it is a global method) and quantify the effect of interactions that can be unnoticed on a local sensitivity analysis approach (i.e., when moving one parameter at a time). Here, we used the Sobol' sensitivity analysis method (Sobol, 2001) and reported 'total order indices' that quantify the contribution of each parameter to the variance of the model output.

Two different physiological phases control the dynamics of plant desiccation under extreme drought, according to whether
$\psi_{LSym}$ is above or below the point of stomatal closure (Fig. 1a). Three time-based metrics were, therefore, considered to explore the sensitivity of plant desiccation to input parameters: (i) the time to hydraulic failure, (ii) the time to stomatal closure, and (iii) the survival time, defined as the time difference between hydraulic failure and stomatal closure (see an illustration of these metrics in Fig. 4). We performed a sensitivity analysis for three different tree species with contrasted ecology and exhibiting various combinations of input parameters (Tab. 3). For each parameter, we randomly sampled a value within a range of $\pm\,20\%$
of the observed value. Starting from a wet soil, and without further precipitation, we ran simulations until hydraulic failure of the plant, defined as the moment when leaves reach 99 % loss of hydraulic conductivity ($PLC_L$>=99 %). This threshold guarantees that plant water pools were almost empty and that no other water reservoirs are available for the plant. Daily climate inputs were set constant according to the simulations shown in Sect. 4. In total, we ran 700,000 simulations in the sensitivity experiment

We based our selection of stand, plant and soil parameters used in the sensitivity analysis on the results from preliminary analyses and from findings by CPRM21. For stand traits, we focused on $LAI_{max}$, canopy conductance ($g_{canopy}$) and total available water content available for the plant ($TAW$). TAW is not an input parameter in *SurEau-Ecos*, but it is an integrative index resulting from the interaction between soil characteristics and rooting depth. To make $TAW$ vary in simulations without affecting soil physical properties, we adjusted rooting depth such as to match the targeted $TAW$. $g_{canopy}$ is also not an input
parameter in *SurEau-Ecos* but also an integrative index resulting from the interactions between $g_{bound}$, $g_{stom\_max}$ and $g_{crown0}$. The following physiological and hydraulic traits were also included: $P_{50}$, $slope_{gs}$, $\psi_{gs,50}$, $slope_{gs}$, $k_{R-SApo,max}$, $k_{SApo-LApo,max}$ $k_{SSym}$, $g_{cuti20}$, $Q_{10a}$ and $Q_{10b}$, $\pi_0$ and $V_S$ (see definition in Tab. 1).

Our results showed that a few parameters explained most of the variability in the resilience of trees to extreme drought (Fig. 4) albeit their importance largely depended on the physiological phase under study. Variations in $LAI_{max}$, $TAW$, $\psi_{gs,50}$ and
$g_{canopy}$ mainly explained the variance in time to stomatal closure, i.e., the first physiological phase. It suggests that, in this phase, interactions between how much water is available in the soil (TAW) and how fast plant transpiration will empty that reservoir ($LAI_{max}$, $\psi_{gs,50}$ and $g_{canopy}$) determine the time to stomatal closure. In the second phase (after stomatal closure), survival time was mostly driven by $LAI$, $g_{min20}$, $Q_{10a}$, $P_{50}$, $TAW$, and $V_S$. In that phase, the importance of stand parameters ($LAI_{max}$ and $TAW$) decreased to the benefit of traits related to the rate of water losses through cuticular transpiration ($g_{cuti20}$





and $Q_{10a}$,), the volume of water reservoirs in the root, trunk and branches ($V_S$) and plant resistance to cavitation ($P_{50}$). When both phases were considered jointly, we observed that the variability in the time to hydraulic failure was mainly associated with stand parameters ($LAI$ and $TAW$) and to a lesser extent with $\psi_{gs,50}$ and $g_{cuti20}$.

We also observed that the patterns described here above were almost identical regardless of the vegetation type under study. In particular, the parameters controlling 'time to hydraulic failure' and 'survival time' were similar among the three studied

vegetation types, suggesting a similarity of plant adaptation strategies to avoid hydraulic failure in a changing climate. The one exception to this pattern is the importance of varying plant resistance to cavitation ($\psi_{50,L}$) in survival time. The influence of $\psi_{50,L}$ ranged from low for *Quercus ilex* (about 0.05) to very important for *Quercus Petraea* (about 0.37). This observation suggests that less drought-resistant species (with higher $\psi_{50,L}$) have a more direct benefit of lowering their $\psi_{50L}$ to increase their survival time than drought-resistant species (with lower $\psi_{50,L}$). This might be due to the non-linear response of water

potential to soil and plant water content, which implies that the rate of change of plant water potential increases as soil and plant water content decreases.



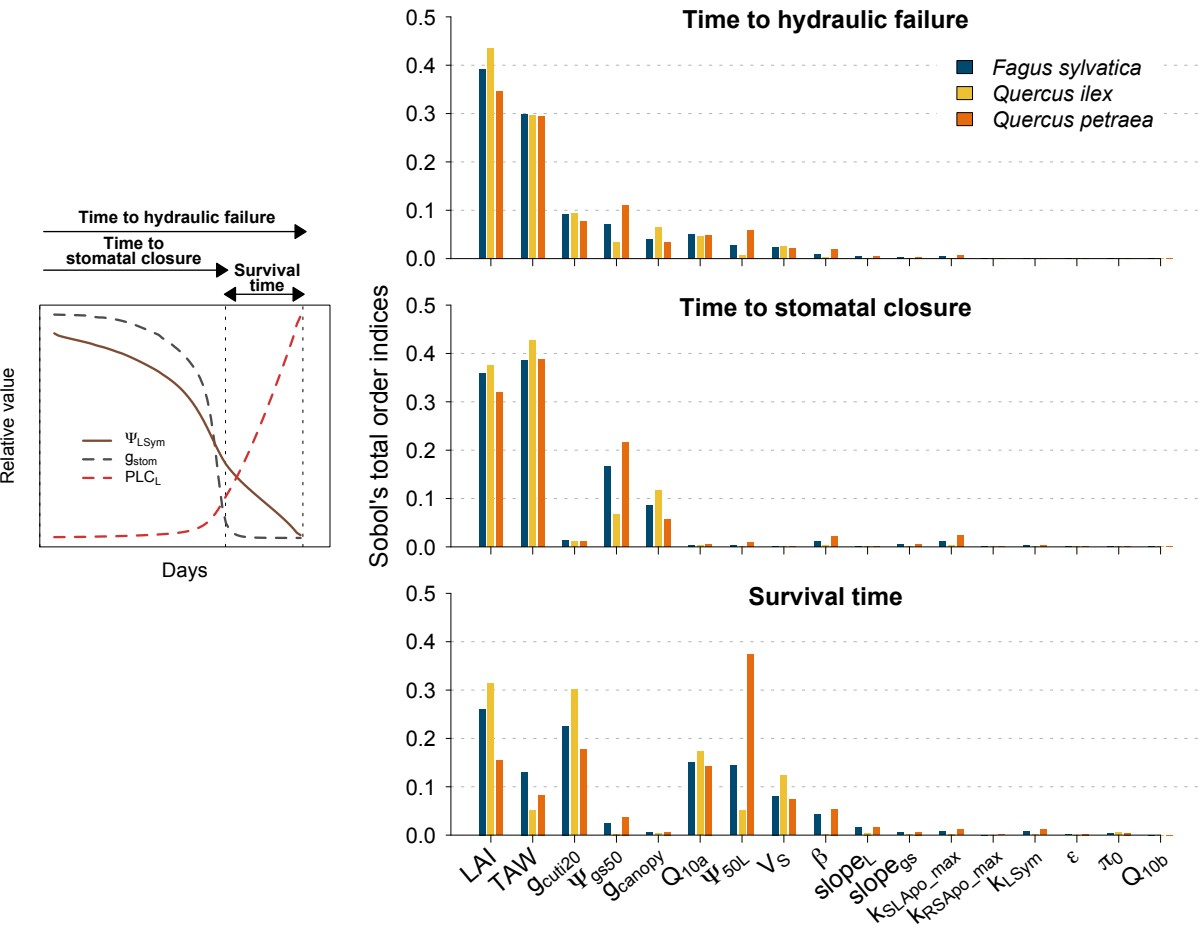

**Fig. 4: Global sensitivity analysis of plant desiccation dynamics to the main hydraulic traits and stand parameters in *SurEau-Ecos*, shown for three different tree species with contrasted ecology and exhibiting various combinations of input parameters (see main input parameters in Tab. 3). We explored the sensitivity of three physiological time-based metrics to input parameters: time to stomatal closure, time to hydraulic failure, and survival time. These three metrics describe the two different physiological phases controlling the dynamics of plant desiccation under extreme drought, according to whether $\psi_{LSym}$ is above or below the point of stomatal closure (Fig. 1a). All hydraulic and stand traits varied from +- 20 % around their original value. TAW is the total available water for the plant.**





Our results shed some light on our understanding of plant functioning under extreme drought. We highlighted the prominent
role of stand traits, namely $LAI$ and $TAW$, along with $\psi_{gs,50}$ in determining the time needed to reach stomatal closure. By
contrast, physiological variables, namely $g_{cuti20}$, $Q_{10a}$, $V_S$ and $\psi_{50,L}$ played a more important role in determining 'survival
time'. Two improvements to the present analyses may strengthen these findings. First, numerous correlations exist between
those traits, reflecting trade-offs and plant functioning strategies (Bartlett et al., 2012; Christoffersen et al., 2016) that we did
not take into account. Second, the relative importance of input parameters is likely to be influenced by climate. For instance,
we would expect the influence of $g_{min20}$, $Q_{10a}$ and $Q_{10b}$ on survival time to increase when temperature increases, following
previous results showing the vulnerability of trees during heatwaves (Cochard et al. 2020). Integrating these potential
improvements in future simulations may further help elucidate the specific spatial and temporal patterns of drought-induced
mortality (Meir et al., 2015).

| Parameter | Quercus ilex | Fagus sylvatica | Quercus Petraea |
|---|---|---|---|
| $LAI_{max}$ | 3 | 5 | 6 |
| $\varepsilon_L$ | 10 | 10 | 10 |
| $\pi_{0_L}$ | -2.1 | -1.9 | -2.1 |
| $\psi_{gs,50}$ | -2.8 | -1.8 | -2.34 |
| $slope_{gs}$ | 44 | 130 | 92 |
| $g_{cuti20}$ | 3 | 4 | 3 |
| $Q_{10a}$ | 1.2 | 1.2 | 1.2 |
| $Q_{10b}$ | 4.8 | 4.8 | 4.8 |
| $T_{Phase}$ | 37.5 | 39 | 42 |
| $\psi_{50,L}$ | -7 | -3.15 | -3.4 |
| $slope_L$ | 30 | 40 | 60 |
| $K_{R-SApo,max}$ | 2.5 | 2.5 | 1.55 |
| $K_{SApo-LApo,max}$ | 5 | 5 | 3.1 |
| $K_{SSym}$ | 0.26 | 0.26 | 0.26 |
| $K_{LSym}$ | 2.5 | 2.5 | 1.55 |
| $g_{stom\_max}$ | 200 | 200 | 200 |
| $\beta$ | 0.97 | 0.98 | 0.97 |
| $V_S$ | 20 | 33 | 40 |


**Tab. 3. Main parameter's values used in model simulations whose results are shown in Figs. 4 and 5. Parameters derived from pressure-volume curves and PLC curve were set equal for the leaf and leaf ($\varepsilon_S = \varepsilon_L$, $\pi_{0_S} = \pi_{0_L}$, $slope_S = slope_L$, $\psi_{50,S} = \psi_{50,L}$).**





## 6. Regional prediction of climate-change impacts on tree mortality

In this section, we aimed to illustrate the potentialities which *SurEau-Ecos* will provide to improve our understanding of forest

vulnerability to drought. We explored whether the probability of plant hydraulic failure simulated by *SurEau-Ecos* was related
to the distribution of two tree species across climate gradients and, then, to identify future areas at risk of drought-induced tree
mortality. Specifically, we hypothesized that hydraulic failure was a significant constraint to tree distribution at the regional
level.

We quantified the probability of hydraulic failure over France (544,000 km$^2$) for two different species chosen for their

contrasted functioning strategies: an evergreen Mediterranean Oak (*Quercus ilex*) and a temperate deciduous European Beech
(*Fagus sylvatica*) (see main parameters in Tab. 3). *Quercus ilex* is a drought-resistant species with low LAI, P$_{50}$, and deep-root
systems to extract water from cracks in the bedrocks during drought. By contrast, *Fagus Sylvatica* is characterized by a higher
vulnerability to drought (higher P$_{50}$) and higher LAI values. As in Sect. 5, we defined hydraulic failure as the point when leaves
reach 99 % loss of hydraulic conductivity ($PLC_L$>=99 %). For each period investigated, we reported the probability of hydraulic

failure as the frequency of years during which $PLC_L$>=99 %.

We ran simulations for present (1991-2020) and future (2071-2100) periods at an 8 km$^2$ resolution over France for both species.
Climate data for the present period (1970-2020) were extracted from the SAFRAN climate reanalysis database (Vidal et al.,
2010) that covers France at an 8 km$^2$ resolution. Projections of climate variables for the future climate period (2071-2100)
were obtained from a climate simulation program involved in the 5$^{th}$ phase of the Coupled Model Intercomparison Project

(CMIP5) and produced as part of the EURO-CORDEX initiative (Kotlarski et al., 2014). One single GCM-RCM couple was
extracted for these analyses (i.e., MPI-ESM-REMO2009), which was chosen because of its averaged climate trajectory over
France when compared to an ensemble of GCM-RCM couples (Fargeon et al., 2020; Ruffault et al., 2020). Data were extracted
at a 0.44° spatial resolution for the historical (1990–2005) and future (2006–2099) periods. Model outputs were bias-corrected
and downscaled at the 8 km$^2$ resolution using a quantile-quantile correction approach (Ruffault et al., 2014).

To apply the model at the landscape scale, we made several simplifying assumptions. First, we assumed that each 8 km$^2$ grid
cell was covered by trees of the same species and LAI was set to a constant value representative of observed values for the
considered species (Tab. 3). Second, soil characteristics were also set constant over the territory. Both assumptions are
unrealistic because stand characteristics vary at the local scale and have a primordial role in the probability of hydraulic failure
(Sect. 5). However, as we aimed to assess the regional (rather than the local) vulnerability of tree species to changes in climate,

we did not expect this to be a main limitation, provided that the results of these simulations be interpreted accordingly to these
assumptions. To assess whether the probability of hydraulic failure was a good proxy of the current southern range of tree
species distribution, we compared the results of our simulations with presence/absence data for each species. Tree species data
were extracted from the national forest inventory database (available at *http://www.ifn.fr*) and aggregated to obtain presence-
absence on the 8-km studied grid, following Cheaib et al. (2012).



Maps of probability of hydraulic failure (probability of reaching $PLC_L >= 99$ %) are shown in Figure 5. We observed contrasted regional patterns according to the species under study. We observed a higher probability of mortality in south-eastern France for both species but the probability of hydraulic failure was higher for the European Beech than for the holm oak. In the rest of the country, the probability of hydraulic failure was almost 0 for the holm oak. By contrast, we observed probabilities up to 50 % for the European beech in the western part and middle of the country, where the climate is temperate. When comparing

these results with the maps of current species distribution, we observed a reasonable degree of spatial agreement between our simulations and presence/absence data. European beech was predominantly present in areas where our simulations indicated a probability of drought-induced mortality equal to 0 %. However, we could not interpret the results for the holm oak in the same way since the current distribution of this species indicates that the southern climate margin is not reached in the present climate.

Our projections for the end of the century showed a future increase in the areas characterized by a high risk of hydraulic failure over France. For *Fagus Sylvatica* the areas characterized by a high risk of hydraulic failure will extent towards the northeast and west of the country over the major part of the territory. For *Quercus ilex*, our simulations indicated that the probability of hydraulic failure should significantly increase in the South-eastern France where this species is currently widespread.

Altogether, these results indicate that future climate conditions might overcome the capacity of the two studied tree species to

face drought over the French territory, which might increase the likelihood of tree mortality and wildfires in the future. Adding information on LAI and soil physical properties might further refine our simulation results. LAI can be estimated from remote sensing indices (see for instance De Kauwe et al., 2020b). However, TAW estimations are more problematic because the information on root depth is rarely available (Ruffault et al., 2013; Venturas et al., 2020).



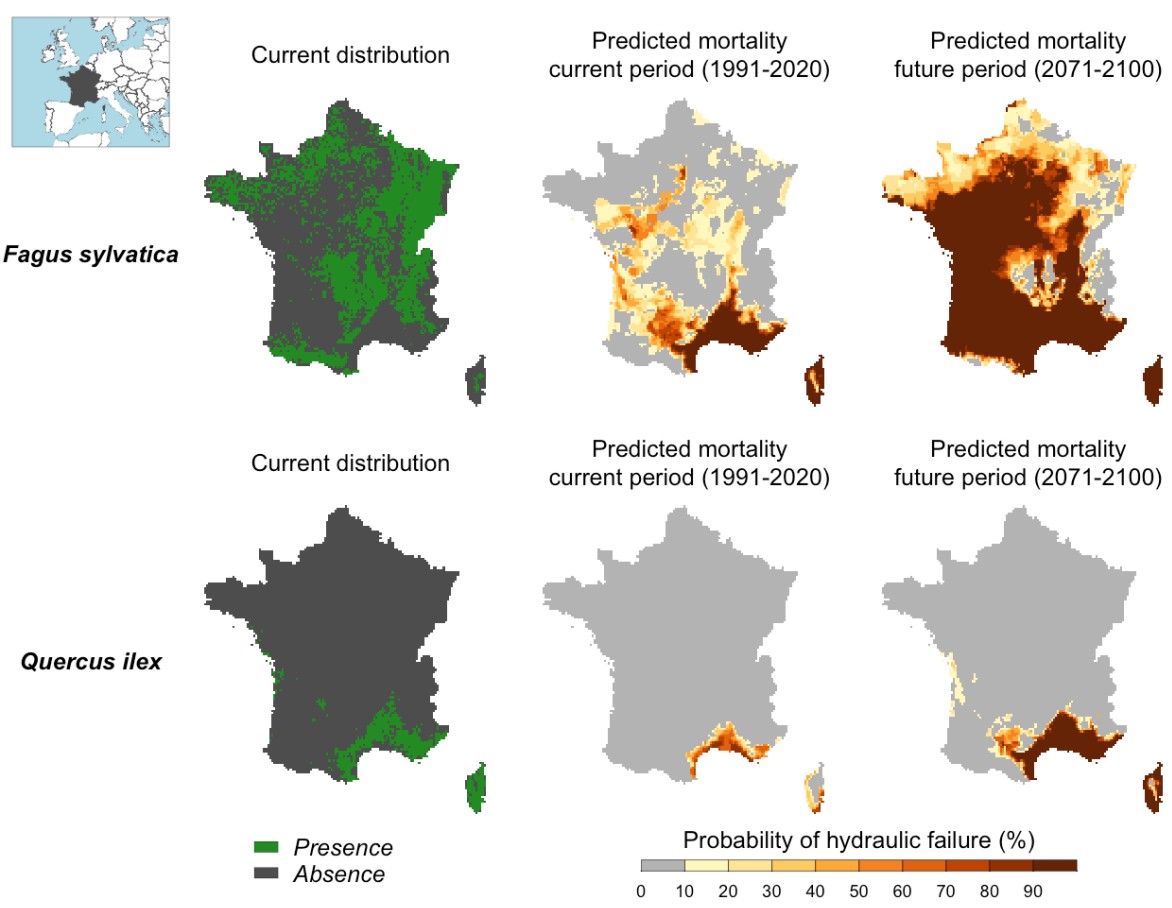

**Fig. 5: Probability of hydraulic failure (%) over the past (1991-2020) and future (20717-2100) period for two tree species in France simulated with *SurEau-Ecos*. The Current distribution is shown for comparison with the simulated risk of hydraulic failure.**

### 7. Conclusion

Drought is arguably one of the most important natural disturbance threatening forest ecosystems in a number of regions worldwide (Anderegg et al. 2020 ). The challenges facing our understanding of the role of plant hydraulics in vegetation dynamics are numerous (McDowell et al., 2019), one being the ability of current vegetation models, including those based on plant hydraulics, to predict plant desiccation dynamics at regional scales (De Kauwe et al., 2020; Rowland et al., 2021; Trugman et al., 2021; Venturas et al., 2020). Here, we presented *SurEau-Ecos*, a new plant hydraulic SPA model aimed at predicting plant water status and drought-induced mortality at scales from stand to region. *SurEau-Ecos* was designed to simulates plant water status of the different plant's compartments, while balancing for the need of input parameters and computational requirements. *SurEau-Ecos* simulates key mechanisms associated with plant desiccation during drought and heatwaves, including the dynamics of plant's water status beyond the point of stomatal closure via residual transpiration flow,





plant cavitation and solicitation of plants' water reservoirs. We showed that *SurEau-Ecos* was able to provide accurate estimations of plant water status dynamics compared to the *SurEau* model, despite that this latter represents plant hydraulics mechanisms in more details. This confirms that, for large-scale applications, the changes we implemented in *SurEau-Ecos*

largely outweigh a potential loss of accuracy associated with the simplification of plant architecture and hydraulic processes. *SurEau-Ecos* provides the capability to better understand the role of plant-hydraulics in vegetation dynamics under climate change conditions characterized by increased drought frequency.

## Appendix A: Leaf phenology module in *SurEau-Ecos*

Leaf area index ($LAI$) of the stand is updated daily. Species can have either evergreen or winter deciduous phenology.

Evergreen species are assumed to maintain a constant $LAI$ throughout the year. $LAI$ values of deciduous plants are adjusted as a function of leaf phenology ($\emptyset$) and the maximum of the stand ($LAI_{max}$) such as:

$$LAI = \emptyset \cdot LAI_{max} \qquad (A1)$$

$\emptyset$ is set to 0 until budburst occurs. Budburst is assumed to be driven by the cumulative effect of forcing temperatures ($R_f$) on bud development (Chuine and Cour, 1999) such as:

$$\sum_{t_0}^{t_f} R_f(T_d) \geq F* \qquad (A2)$$

Where $t_0$ is a parameter defining the initial date of the forcing period, $t_f$ the budburst date and F* is a parameter defining the amount of forcing temperature to reach budburst. Once budburst, $\emptyset$ increases from 0 to 1 at a rate specified by a parameter describing the LAI growth rate per day ($R_{LAI}$). In autumn, leaf fall occurs ($\emptyset$ starts to decline) when the average daily temperature falls below 5ºC (De Cáceres et al., 2015; Sitch et al., 2003) and then $\emptyset$ declines at a similar rate to LAI growth in

spring.

## Appendix B: additional tables and figures

| Symbol | Unit | Description |
|---|---|---|
| $T_{mean}$ | °C | Mean temperature |
| $T_{min}$ | °C | Minimum temperature |
| $T_{max}$ | °C | Maximum temperature |
| $R_{global}$ | MJ.m$^{-2}$ | Global radiation |
| $ppt$ | mm | Precipitation |
| $RH_{mean}$ | % | Mean relative humidity |
| $RH_{min}$ | % | Minimum relative humidity |
| $RH_{max}$ | % | Maximum relative humidity |
| $u$ | m.s$^{-1}$ | Mean wind speed |





**Tab. B1**: Daily climate input variables.


| | | | *SurEau* "Trunk-only" | | |
|---|---|---|---|---|---|
| | **Parameters** | **Leaf** | **Branches** | **Trunk** | **Root** |
| Symplasm | $\pi_0$ (MPa) | 2.1 | -2.1 | -2.1 | -2.1 |
| | $\epsilon$ (MPa$^{-1}$) | 10 | 10 | 10 | 10 |
| | $K$ (mmol s$^{-1}$ MPa$^{-1}$.m$^{-2}$) | 1.80 | 0.55 | 0.26 | 7.22 |
| | $Qsat$ (mol.m$^{-2}$) | 43.75 | 91.12 | 355.73 | 377.77 |
| | Surface (m$^2$) | 10.5 | 5.8 | 2.7 | 54.1 |
| Apoplasm | *P50* (MPa) | -3.4 | -3.4 | -3.4 | -3.4 |
| | *Slope* (% MPa$^{-1}$) | 60 | 60 | 60 | 60. |
| | $K$ (mmol s$^{-1}$ MPa$^{-1}$.m$^{-2}$) | 1.32 | 6.50 | 7.16 | 2.03 |
| | $Qsat$ (mol/m$^{-2}$) | 14.58 | 182.25 | 711.46 | 658.8 |
| | | | *SurEau-Ecos* | | |
| | **Parameters** | **Leaf** | **Stem** | | |
| Symplasm | $\pi_0$ (MPa) | -2.1 | -2.1 | | |
| | $\epsilon$ (MPa$^{-1}$) | 10 | 10 | | |
| | $K$ (mmol s$^{-1}$ MPa$^{-1}$.m$^{-2}$) | 1.80 | 0.84 | | |
| | $Qsat$ (mol.m$^{-2}$) | 4.16 | 78.53 | | |
| | Surface (m$^2$) | 10.5 | 62.6 (All wood) | | |
| Apoplasm | *P50* (MPa) | -3.4 | -3.4 | | |
| | *Slope* (% MPa$^{-1}$) | 60 | 60 | | |
| | $K$ (mmol s$^{-1}$ MPa$^{-1}$.m$^{-2}$) | 1.32 | 3.4 | | |
| | $Qsat$ (mol/m$^{-2}$) | 1.4 | 148 | | |

**Tab. B2 Main physiological parameters of plant compartments used for the comparison between *SurEau* and *SurEau-Ecos*.**




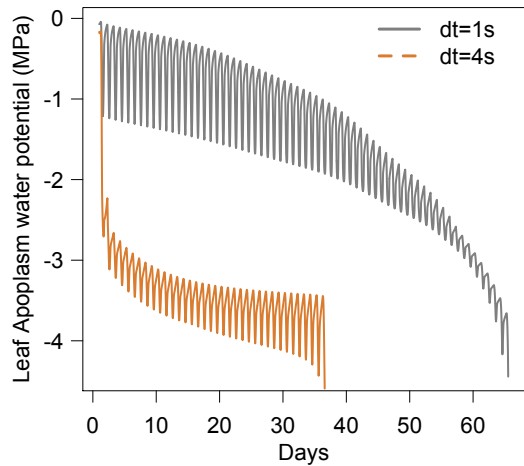

**Fig. B1: Illustration of the constraint on *dt* due to the Courant–Friedrichs–Lewy (CFL) condition in *SurEau-Ecos*. When dt > 2*K/C numerical instabilities are observed.**



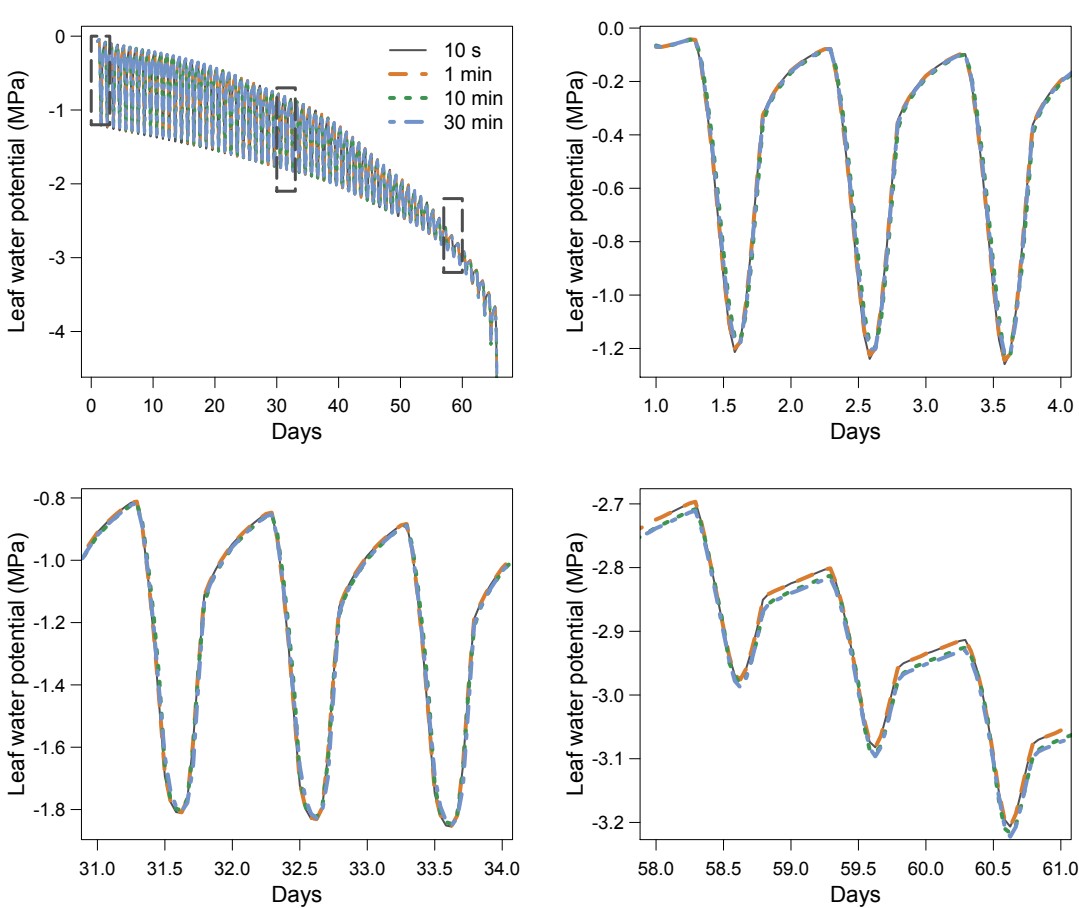

**Fig. B2: Impact of the time step (dt) on simulation results with the 'implicit' resolution scheme.**



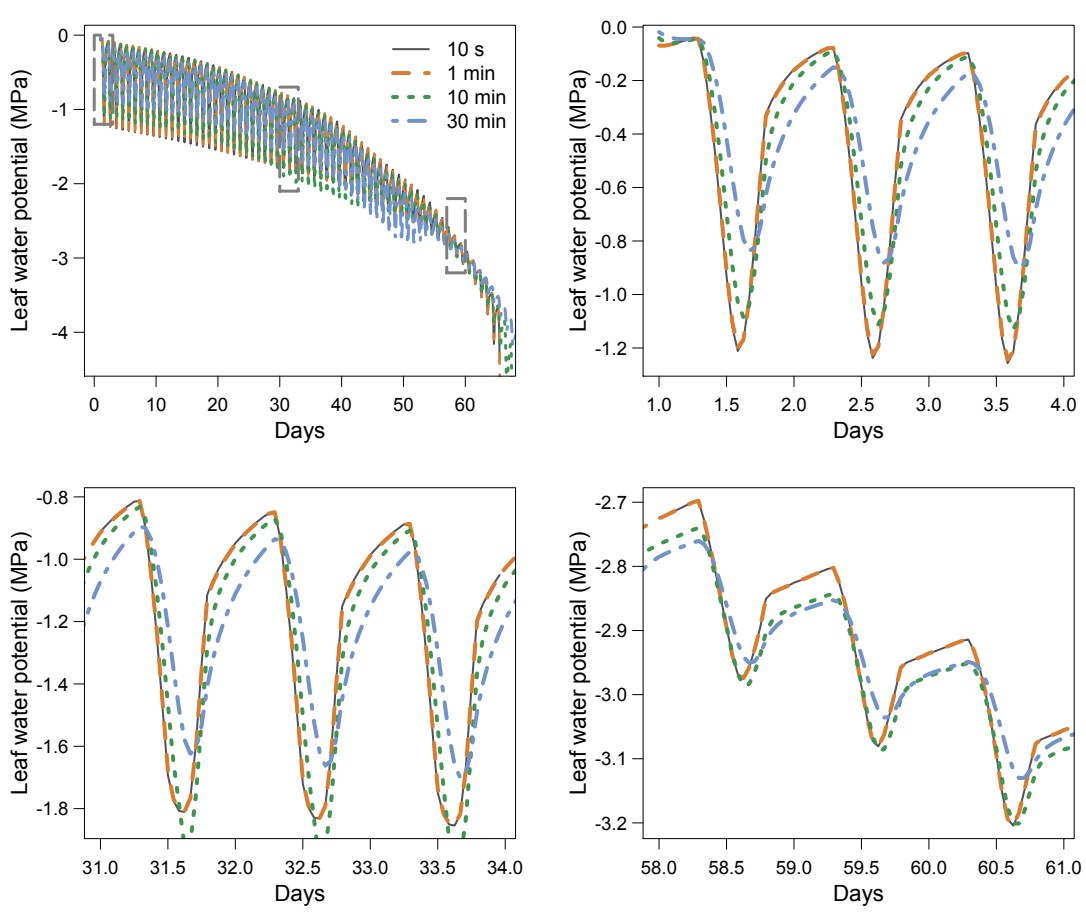

**Fig. B3: Impact of the time step (dt) on simulation results with the 'semi-implicit' resolution scheme.**




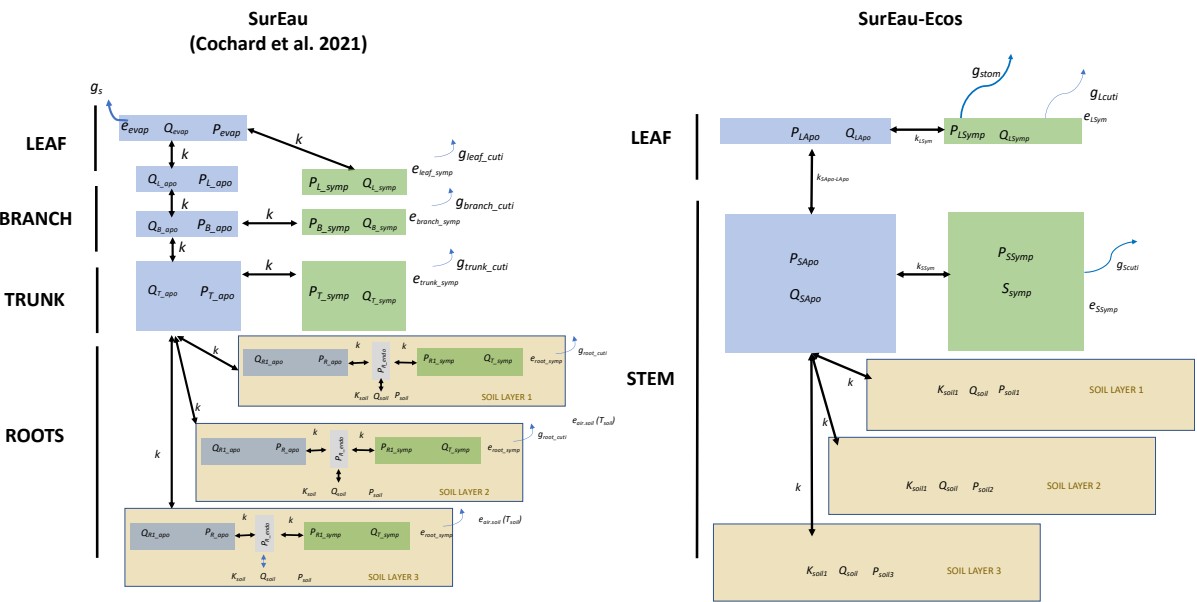

**Fig. B4: Comparison of the plant architecture in *SurEau* and *SurEau-Ecos*. Q indicates the water quantities of the compartments, P the water potentials, k the hydraulic conductances, gs the gaseous stomatal conductances and g$_{cuti}$ the gaseous cuticular conductances. The subscripts *Apo*, *Sym* and *Endo* indicate the apoplasm, symplasm and endoderm compartments, respectively. The subscripts L, S, B, T, and evap stand for Leaf, Stem, Branch, Trunk, Root and Evaporative site respectively**



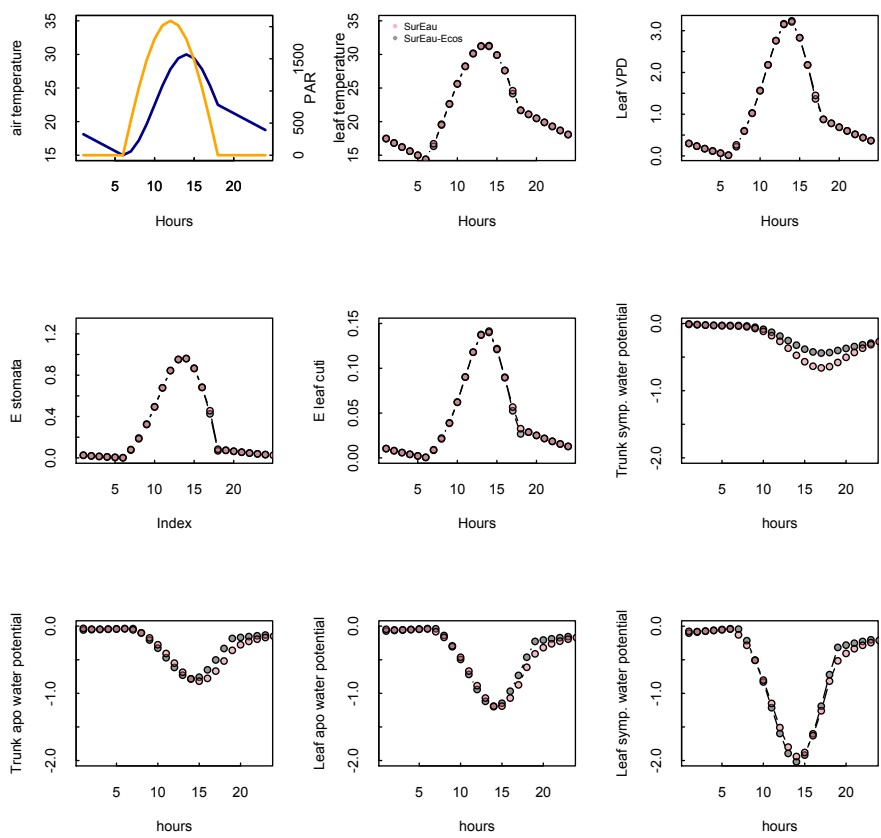

**Fig. B5: Comparison of hourly outputs between *SurEau* and *SurEau-Ecos* for the first day of simulation. Climatic inputs (Radiation, temperature and VPD) are shown in the first panel.**





**Fig. B6: Comparison of the water potential and the water discharge dynamics for the first day of simulation between the Trunk**
**compartment of *SurEau* and the Stem compartment of *SurEau-Ecos* for two different parameterizations. In the upper panels**
***SurEau-Ecos* was parameterized with the stem symplasmic water volume computed as the sum of the symplasmic water volumes of**
**the roots, trunk and branches of *SurEau*. In lower panels, the stem symplasmic water volume of SurEau-Ecos was considered to be**
**equivalent to the trunk water volume of *SurEau* (see details in Tab. B2).**





**Appendix C: numerical schemes**

**C1. Explicit scheme**

Let $\delta_L^{cav} = \begin{cases} 0, & if \ \psi_{LApo}^n \geq \psi_{LApo}^{mem} \ (no \ cavitation \ event) \\ 1, & if \ \psi_{LApo}^n < \psi_{LApo}^{mem} \ (cavitation \ event) \end{cases}$

And

Let $\delta_S^{cav} = \begin{cases} 0, & if \ \psi_{SApo}^n \geq \psi_{SApo}^{mem} \ (no \ cavitation \ event) \\ 1, & if \ \psi_{SApo}^n < \psi_{SApo}^{mem} \ (cavitation \ event) \end{cases}$


Applying the explicit scheme (Eq. 48 in main text) to the four water balance equations (Eq. 6 to 9 in main text):

Eq. 6 can be rearranged to determine $\psi_{LApo}^{n+1}$:

$$\psi_{LApo}^{n+1} = \psi_{LApo}^n + \frac{dt}{C_{LApo}}\left(K_{SLApo}\left(\psi_{SApo}^n - \psi_{LApo}^n\right) + k_{LSym}\left(\psi_{LSym}^n - \psi_{LApo}^n\right) + \delta_L^{cav}K_L^{cav}\left(\psi_{LApo}^{cav} - \psi_{LApo}^n\right)\right) \qquad \text{C1.1}$$

Similarly, Eq. 7 gives $\psi_{SApo}^{n+1}$:

$$\psi_{SApo}^{n+1} = \psi_{SApo}^n + \frac{dt}{C_{SApo}}\left(K_{SLApo}\left(\psi_{LApo}^n - \psi_{SApo}^n\right) + K_{SSym}\left(\psi_{SSym}^n - \psi_{SApo}^n\right) + \delta_S^{cav}K_S^{cav}\left(\psi_{SApo}^{cav} - \psi_{SApo}^n\right) \right.$$
$$\left. + \sum_j K_{soil_jS}\left(\psi_{soil_j}^n - \psi_{SApo}^n\right)\right) \qquad \text{C1.2}$$

Then Eq. 8 and 9 give $\psi_{LSym}^{n+1}$ and $\psi_{TSym}^{n+1}$

$$\psi_{LSym}^{n+1} = \psi_{LSym}^n + \frac{dt}{C_{LSym}}\left(K_{LSym}\left(\psi_{LApo}^n - \psi_{LSym}^n\right) - Estom^{n+\frac{1}{2}} - Ecuti_L^{n+\frac{1}{2}}\right) \qquad \text{C1.3}$$

$$\psi_{SSym}^{n+1} = \psi_{SSym}^n + \frac{dt}{C_{SSym}}\left(K_{SSym}\left(\psi_{SApo}^n - \psi_{SSym}^n\right) - Ecuti_S^{n+\frac{1}{2}}\right) \qquad \text{C1.4}$$






## C2. Implicit scheme

By combining equations 6, 7, 8 and 9 with the implicit discretization (50), it is possible to analytically compute the unknown water potentials of each compartment at time $n+1$.

First, we eliminate $\psi_{LSym}^{n+1}$ in equations 6 and 8 by summing $(6) + (8) \times \dfrac{K_{LSym}}{K_{LSym} + \frac{C_{LSym}}{dt} + \frac{E'^{n+\frac{1}{2}}}{2}}$ and re-organizing:

$$
\frac{C_{LApo}}{dt}\left(\psi_{LApo}^{n+1} - \psi_{LApo}^{n}\right) + K_{SLApo}\left(\psi_{LApo}^{n+1} - \psi_{SApo}^{n+1}\right) + \frac{K_{LSym}\left(\frac{C_{LSym}}{dt} + \frac{E'^{n+\frac{1}{2}}}{2}\right)}{K_{LSym} + \frac{C_{LSym}}{dt} + \frac{E'^{n+\frac{1}{2}}}{2}}\left(\psi_{LApo}^{n+1} - \psi_{LSym}^{n}\right)
$$
$$
+ \delta_L^{cav} K_L^{cav}\left(\psi_{LApo}^{n+1} - \psi_{LApo}^{cav}\right) + \frac{K_{LSym}}{K_{LSym} + \frac{C_{LSym}}{dt} + \frac{E'^{n+\frac{1}{2}}}{2}}\left(E^{n+\frac{1}{2}} + Emin_L^{n+\frac{1}{2}}\right) = 0
$$

C2.1

With $\delta_L^{cav} = \begin{cases} 0, & if\ \psi_{LApo}^{n+1} \geq \psi_{LApo}^{mem}\ (no\ new\ cavitation\ event) \\ 1, & if\ \psi_{LApo}^{n+1} < \psi_{LApo}^{mem}\ (new\ cavitation\ event) \end{cases}$

Let define intermediate variables to ease the resolution:

$$
\widetilde{K_L} = \frac{C_{LApo}}{dt} + \frac{K_{LSym}\left(\frac{C_{LSym}}{dt} + \frac{E'^{n+\frac{1}{2}}}{2}\right)}{K_{LSym} + \frac{C_{LSym}}{dt} + \frac{E'^{n+\frac{1}{2}}}{2}} + \delta_L^{cav} K_L^{cav}
$$

C2.2

and

$$
\widetilde{\psi_L} = \frac{\frac{C_{LApo}}{dt}\psi_{LApo}^{n} + \frac{K_{LSym}\left(\frac{C_{LSym}}{dt} + \frac{E'^{n+\frac{1}{2}}}{2}\right)}{K_{LSym} + \frac{C_{LSym}}{dt} + \frac{E'^{n+\frac{1}{2}}}{2}}\psi_{LSym}^{n} + \delta_L^{cav} K_L^{cav}\psi_{LApo}^{cav}}{\widetilde{K_L}}
$$

C2.3

and

$$
\widetilde{E_L} = \frac{K_{LSym}}{K_{LSym} + \frac{C_{LSym}}{dt} + \frac{E'^{n+\frac{1}{2}}}{2}}\left(Estom^{n+\frac{1}{2}} + Ecuti_L^{n+\frac{1}{2}}\right)
$$

C2.4


Now, equation C2.1 can be rewritten:

$$
\widetilde{K_L}\left(\psi_{LApo}^{n+1} - \widetilde{\psi_L}\right) + K_{SLApo}\left(\psi_{LApo}^{n+1} - \psi_{SApo}^{n+1}\right) + \widetilde{E_L} = 0
$$

C2.5



Similarly, eliminating $\psi_{SSym}^{n+1}$ in equations 7 and 9 by summing $(7) + (9) \frac{1}{1+\frac{C_{SSym}}{k_{SSym}\,dt}}$ and re-organizing, this leads to:

$$\frac{C_{SApo}}{dt}\left(\psi_{SApo}^{n+1} - \psi_{SApo}^n\right) + K_{SLApo}\left(\psi_{SApo}^{n+1} - \psi_{LApo}^{n+1}\right) + \sum_j K_{soil_jS}\left(\psi_{SApo}^{n+1} - \psi_{soil_j}^n\right) + \frac{1}{\frac{1}{K_{SSym}} + \frac{dt}{C_{SSym}}}\left(\psi_{SApo}^{n+1} - \psi_{SSym}^n\right)$$

$$+ \delta_S^{cav} K_S^{cav}\left(\psi_{Sapo}^{n+1} - \psi_{SApo}^{cav}\right) + \frac{Ecuti_S^{n+\frac{1}{2}}}{1 + \frac{C_{SSym}}{K_{SSym}\,dt}} = 0 \qquad \text{C2.6}$$

With $\delta_S^{cav} = \begin{cases} 0, & \text{if } \psi_{SApo}^{n+1} \geq \psi_{SApo}^{cav}\ (\text{no new cavitation event}) \\ 1, & \text{if } \psi_{SApo}^{n+1} < \psi_{SApo}^{cav}\ (\text{new cavitation event}) \end{cases}$

Similarly, defining:

$$\widetilde{K_S} = \frac{C_{SApo}}{dt} + \frac{1}{\frac{1}{K_{SSym}} + \frac{dt}{C_{SSym}}} + \sum_j K_{soil_jS} + \delta_S^{cav} K_S^{cav} \qquad \text{C2.7}$$

and

$$\widetilde{\psi_S} = \frac{\frac{C_{SApo}}{dt}\psi_{SApo}^n + \frac{1}{\frac{1}{K_{SSym}} + \frac{dt}{C_{SSym}}}\psi_{SSym}^n + \sum_j K_{soil_jS}\,\psi_{soil_j} + \delta_S^{cav} K_S^{cav}\psi_{SApo}^{cav}}{\widetilde{K_S}} \qquad \text{C2.8}$$

Equation C2.6 can be rewritten:

$$\widetilde{K_S}\left(\psi_{SApo}^{n+1} - \widetilde{\psi_S}\right) + K_{SLApo}\left(\psi_{SApo}^{n+1} - \psi_{LApo}^{n+1}\right) + \frac{Ecuti_S^{n+\frac{1}{2}}}{1 + \frac{C_{SSym}}{K_{SSym}\,dt}} = 0 \qquad \text{C2.9}$$

Now, we eliminate $\psi_{SApo}^{n+1}$ from simplified equations C2.5 and C.9 by summing $(C5) + (C9) \times \frac{K_{SLApo}}{K_{SLApo} + \widetilde{K}}$ and re-organizing:

$$\widetilde{K_L}\left(\psi_{LApo}^{n+1} - \widetilde{\psi_L}\right) + \frac{K_{SLApo}\widetilde{K_S}}{K_{SLApo} + \widetilde{K_S}}\left(\psi_{LApo}^{n+1} - \widetilde{\psi_S}\right) + \widetilde{E_L} + \frac{K_{SLApo}}{K_{SLApo} + \widetilde{K_S}}\frac{Ecuti_S^{n+\frac{1}{2}}}{1 + \frac{C_{SSym}}{K_{SSym}\,dt}} = 0$$

Let

$$\widetilde{E_S} = \frac{K_{SLApo}}{K_{SLApo} + \widetilde{K_S}}\frac{Ecuti_S^{n+\frac{1}{2}}}{1 + \frac{C_{SSym}}{K_{SSym}\,dt}} \qquad \text{C2.10}$$


These equations can be combined to determine $\psi_{LApo}^{n+1}$, $\psi_{SApo}^{n+1}$, $\psi_{LSym}^{n+1}$ and $\psi_{SSym}^{n+1}$




We can be rearranged to determine $\psi_{LApo}^{n+1}$:

$$\psi_{LApo}^{n+1} = \frac{\frac{K_{SLApo}\widetilde{K_S}}{K_{SLApo}+\widetilde{K_S}}\widetilde{\psi_S} + \widetilde{K_L}\widetilde{\psi_L} - [\widetilde{E_L}+\widetilde{E_S}]}{\frac{K_{SLApo}\widetilde{K_S}}{K_{SLApo}+\widetilde{K_S}} + \widetilde{K_L}}$$

C2.11

Knowing $\psi_{LApo}^{n+1}$, we can determine $\psi_{SApo}^{n+1}$ from Equation C2.5:

$$\psi_{SApo}^{n+1} = \frac{(\widetilde{K_L}+K_{SLApo})\psi_{LApo}^{n+1} - \widetilde{K_L}\widetilde{\psi_L} + \widetilde{E_L}}{K_{SLApo}}$$

C2.12

In practice, because we don't know whether new cavitation events will occur during the time step, equations C2.2 and C2.3 and C2.7 and C2.8 are first computed assuming that $\delta_L^{cav}$ and $\delta_S^{cav}$ did not change since the last time step. This will be correct for most time steps, except those when cavitation either starts or ends. At this stage, we should hence check whether solutions $\psi_{LApo}^{n+1}$ and $\psi_{SApo}^{n+1}$ are below or above $\psi_{LApo}^{cav}$ and $\psi_{SApo}^{cav}$, in order to eventually update $\delta_L^{cav}$ or $\delta_S^{cav}$ if needed. In case of change (for time steps exactly corresponding to begin or end of cavitation events), the computation should be done again with actualized values of $\delta_L^{cav}$ and $\delta_S^{cav}$.

Finally, knowing $\psi_{LApo}^{n+1}$, we can solve $\psi_{LSym}^{n+1}$ from Equation 8:

$$\psi_{LSym}^{n+1} = \frac{K_{LSym}\psi_{LApo}^{n+1} + \left(\frac{C_{LSym}}{dt}+\frac{E'^{n+\frac{1}{2}}}{2}\right)\psi_{LSym}^n - Estom^{n+\frac{1}{2}} - Ecuti_L^{n+\frac{1}{2}}}{K_{LSym}+\frac{C_{LSym}}{dt}+\frac{E'^{n+\frac{1}{2}}}{2}}$$

C2.13

And knowing $\psi_{SApo}^{n+1}$, we can solve $\psi_{LSym}^{n+1}$ from Equation 9:

$$\psi_{SSym}^{n+1} = \frac{K_{SSym}\psi_{SApo}^{n+1} + \frac{C_{SSym}}{dt}\psi_{SSym}^n - Ecuti_S^{n+\frac{1}{2}}}{K_{SSym}+\frac{C_{SSym}}{dt}}$$

C2.14

**C3. Semi-Implicit scheme**

Let $\delta_L^{cav} = \begin{cases} 0, & if\ \psi_{LApo}^{n+1} \geq \psi_{LApo}^{mem}\ (no\ cavitation\ event) \\ 1, & if\ \psi_{LApo}^{n+1} < \psi_{LApo}^{mem}\ (cavitation\ event) \end{cases}$

And

With $\delta_S^{cav} = \begin{cases} 0, & if\ \psi_{SApo}^{n+1} \geq \psi_{SApo}^{cav}\ (no\ new\ cavitation\ event) \\ 1, & if\ \psi_{SApo}^{n+1} < \psi_{SApo}^{cav}\ (new\ cavitation\ event) \end{cases}$

By combining equations 6-9 with the semi-implicit solving (57), it leads to

For $\psi_{LApo}^{n+1}$,





$$\psi_{LApo}^{n+1} = \alpha_{LApo}\psi_{LApo}^n + (1 - \alpha_{LApo})\tilde{\psi}_{LApo}$$

$$\alpha_{LApo} = e^{-\frac{K_{TLApo}+K_{LSym}+\delta_L^{cav}k_L^{cav}}{C_{LApo}}dt} \quad and \quad \tilde{\psi}_{LApo} = \frac{K_{SLApo}\psi_{SApo}^n + K_{LSym}\psi_{LSym}^n + \delta_L^{cav}k_L^{cav}\psi_{LApo}^{cav}}{K_{SLApo} + k_{LSym} + \delta_L^{cav}k_L^{cav}}$$

C3.1

For $\psi_{SApo}^{n+1}$,

$$\psi_{SApo}^{n+1} = \alpha_{SApo}\psi_{SApo}^n + (1 - \alpha_{SApo})\tilde{\psi}_{SApo}$$

$$with \quad \alpha_{SApo} = e^{-\frac{K_{SL}+K_{SSym}+\sum_j K_{soil_jS}+\delta_S^{cav}K_S^{cav}}{C_{SApo}}dt}$$

$$and \quad \tilde{\psi}_{SApo} = \frac{K_{SLApo}\psi_{LApo}^n + K_{SSym}\psi_{SSym}^n + \sum_j K_{soil_jS}\psi_{soil_j}^n + \delta_S^{cav}K_S^{cav}\psi_{SApo}^{cav}}{K_{SLApo} + K_{SSym} + \sum_j K_{soil_jS} + \delta_S^{cav}K_S^{cav}}$$

C3.2

For $\psi_{LSym}^{n+1}$,

$$\psi_{LSym}^{n+1} = \alpha_{LSym}\psi_{LSym}^n + (1 - \alpha_{LSym})\tilde{\psi}_{LSym}$$

$$\alpha_{LSym} = e^{-\frac{k_{LSym}}{C_{LSym}}dt} \quad and \quad \tilde{\psi}_{LSym} = \frac{K_{LSym}\psi_{LApo}^n - Estom^n - Ecuti_L^n}{K_{LSym}}$$

C3.3

For $\psi_{SSym}^{n+1}$,

$$\psi_{SSym}^{n+1} = \alpha_{SSym}\psi_{SSym}^n + (1 - \alpha_{SSym})\tilde{\psi}_{SSym}$$

$$\alpha_{SSym} = e^{-\frac{K_{SSym}}{C_{SSym}}dt} \quad and \quad \tilde{\psi}_{SSym} = \frac{K_{SSym}\psi_{SApo}^n - Ecuti_S^n}{K_{SSym}}$$

C3.4

**Code availability**

The model code along with instructions on how to run the model version presented in this paper are available from https://doi.org/10.5281/zenodo.5878978

**Author Contribution**

JR led the writing of the manuscript with input from all authors. JR and NM coordinated the project. HC and JLD supervised the project. JR, NM, and FP developed the code and conducted the experiments. NM developed a preliminary version of the code. FP designed the numerical resolutions of the model with inputs from NM. All authors read and approved the manuscript.





**Competing interests**

The authors declare that they have no conflict of interest

**Acknowledgments**

This study was funded by the ANR projects 16-IDEX-0001and 18-CE20-0005. JR received funding from ECODIV department
of INRAE

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
