# Peer review of "SurEau-Ecos v2.0: A trait-based plant hydraulics model for simulations of plant water status and drought-induced mortality at the ecosystem level"

_Geoscientific Model Development, 2022_

## Author Comment (AC1)

This manuscript presents a soil-plant-atmosphere transfer model suited for simulating plant dessication and drought-induced mortality. Few models can simulate plant dessication after stomatal closure, and as far as I know SurEau is probably the best option for this purpose. In this respect, bringing SurEau to regional applications by lessening computational burden and simplifying parameter estimation is a good contribution of this paper. Furthermore, the "implicit" numerical scheme can be helpful for other models with similar design of plant architecture. The model presentation is very complete, and I agree that the comparison with the original SurEau can be taken as a sort of model evaluation. I particularly enjoyed the global sensitivity analysis, which nicely illustrates the importance of different plant traits before and after stomatal closure.

Even if the model already constitutes a valuable contribution, there are some points that could be improved. First, I think the authors could have complemented the presentation of the model by discussing how easy is to determine parameter values for multiple species. SurEauEcos decreases the number of parameters with respect to SurEau, but still there are several hydraulic parameters that may be hard to get for most species. In addition, if the model is to be used at the regional scale and for climate change impacts, the process of conduit refilling or replacing via sapwood growth should be somehow accounted for, or at least discussed in the manuscript, since this would overcome the assumption of setting PLC to zero each new year (as the authors did in the application example). Given the importance of LAI both before and after stomatal closure, further refinement of applications could include not only from estimation of spatial LAI variation, but also from coupling SurEauEcos with a model of forest dynamics so that temporal variation of LAI could occur, to better represent the adaptive capacity of forest to climatic changes. Finally, the approach to model soil evaporation (i.e. the minimum of the two supply functions) should be better justified.

We thank the reviewer for this positive appreciation of our work and his thorough revision of the manuscript.

The reviewer highlighted important aspects of the current version of SurEau-Ecos regarding its parametrization. A similar question was also raised by the other reviewer, and we have tried to reinforce our manuscript by providing more elements regarding model representation, the degree of importance of each parameter in the model and how each parameter can be can they be extracted or estimated. One of the main reasons why we developed SurEau-Ecos while SurEau was already available (Cochard et al., 2021) is that we aimed for a different balancing between plant representation in one hand and the possibility to apply the model for operative large-scale purpose on the other. This was achieved through two main changes: (i) implementing different (faster) numerical schemes and (ii) lowering the number of parameters. As consequence, SurEau-Ecos requires fewer parameters than SurEau, mostly thanks to the reduction of the number of plant compartments (removing roots and branches). As noticed by the reviewer, some parameters that may appear hard to find, particularly because they are not commonly used in the ecosystem modelling community. The vast majority of these parameters, however, can be either extracted from available datasets or, when not directly available, be easily derived from these datasets with the proper methodology. To address this important point, we added a section in the manuscript that specifically focus on how to parameterize SurEau-Ecos, including a table, that summarizes, for the most sensitive plant parameters, (i) the level of organization the parameter applies to (soil, leaf, stem, plant or stand), (ii) if it can be readily be extracted from a database, (iii) some potential databases or reference where they can be find, (iv) or how to derive the parameter from data (if not directly available). In addition, for the purpose of predicting hydraulic failure, not all parameters are equally sensitive and some of them can be set to default values if not available. We also provided an index of sensitivity that helps to identify the most critical parameters.

We also agree that the model, in its current form, is mostly applicable at the seasonal drought scale. The processes related to photosynthesis, respiration, growth and carbon allocation that are necessary to account for legacy effects of drought or acclimation have been overlooked there. Such processes are indeed often the focus of most models. When developing *SurEau-Ecos* we envisioned two main types of applications:

- first it could be applied alone, in its current form. This can be useful to estimate spatialized index of vulnerability that could account for both stand level parameters such as leaf area index (derived from remote sensing), soil properties (derived from databases), and species-specific hydraulic traits. In this case it can be forced by remote sensing data and global re-analysis data in order to predict indices such as hydraulic failure and drought survival or moisture content, but it would neglect long term effects and species interactions within a community.
- Alternatively, it could provide a comprehensive hydraulic basis for larger scale land surface, ecosystem or community models. Current projects of the group aim at integrating *SurEau-Ecos* with the forest growth models CASTANEA (Dufrêne *et al.*, 2005) and GO+ (Moreaux et al. 2020) and the gap model ForCEEPS (Morin *et al.*, 2021) under the Capsis platform (Dufour-Kowalski *et al.*, 2012). Thus, future researches and development should focus on how to link carbon and growth metabolism to hydraulic properties and how to model feedbacks between growth and hydraulic properties.

We added a last section to our manuscript providing potential for application of *SurEau-Ecos*, including the current limitations and explaining the possibility to parameterize and to integrate the model into larger scale models.

Finally, regarding our approach to model soil evaporation, we run a few tests to see if integrating the minimum of two functions (the first as a function of PET and the second of VPD) did indeed improve our estimations of the dynamics of soil water content by comparison with the soil water content measured in the 20 first centimeters at the Fontblanche study site. After examination of these results (no shown, we concluded that adding PET to this formulation did not permit to improve our estimations of soil water content in the first soil layer compared to observations. We therefore decided to adopt the more standard formulation such as  $E_{soil}$  depends on the maximum soil conductance ( $g_{soil0}$ ) and the REW of the first soil layer:

$$E_{soil} = g_{soil0}.REW_1.\frac{VPD}{P_{Atm}}$$

We hope that these changes in the manuscript will answer the reviewer's comments and are, of course, prepared to reconsider any point that would remain unclear. Please find below a point-by-point response to the other minor comments raised by the reviewer.

**Minor corrections**

L11. In some parts of the ms, the model is referred to as 'plant hydraulic model' and in others as a 'soil-plant-atmosphere (SPA) model'. Please homogenize.

Thank you for this comment. As the reviewer noticed, *SurEau-Ecos* is both a plant hydraulic model and a SPA model. We agree with the reviewer that switching from one form of expression to another is likely to create confusion for the reader. For consistency, we now referred to *SurEau-Ecos* as a plant hydraulic model and kept that definition throughout the manuscript.

L19. 'schemes'

Corrected

L45. The acronym 'SPA' has not yet been defined.

Corrected. The term SPA has been removed from this sentence and is now introduced in the next paragraph.

Fig. 1. I suggest moving the rectangle 'soil water balance' into the upper box (stand water balance), since it does not strictly belong to plant hydraulics. Alternatively, change the labels of the two boxes.

We thank the reviewer for this relevant comment. We agree that soil water balance should not be considered as "plant hydraulic" process, but nor can it be integrated with the "stand water balance box" as its temporal resolution is that of plant hydraulics (1-3600s). The best option was to change the labels of the boxes. The first box that represent the process at a daily time step is now called "stand water balance" and the second box that represents processes at a smaller time step is called 'Plant hydraulics and soil water balance".

L112. "To account for..." the sentence has no verb. Revise.

Corrected.

L116. Notation: 'Q' or 'q'? Similarly 'S' or 's'? In eq. (1) these letters were in lower case.

We thank the reviewer for rising this point and apologize for these unclear notations and units in the manuscript. Q (kg.m-2leaf) results from the volumetric integration of q (kg.m-3). Please note here that, for convenience, the state variable Q is expressed per unit leaf area. We added these different units in the manuscription to help to clarify these points.

L134. 'controls'

Corrected

L136. 'units'

Corrected.

L149. It would be nice to specify the code availability, here or somewhere in the ms.

The model code along with instructions on how to run the current version of the model are available from https://doi.org/10.5281/zenodo.5878978. This is specified in the section "code availability" at the end of the manuscript. The most recent version of the code is also available on GitHub from https://github.com/julien-ruffault/SurEau-Ecos

L161. Remove 'by'

Corrected

L185. 'The third term represents...' (no fourth term here)

Corrected

L231. KRjT? Shouldn't it be K\_Rj-Sapo?

Yes, Corrected

Eq. 25. Remove right-hand '='

Corrected

L256. 'E\_leaf' or 'E\_L'?

**Corrected**

Eq. 33. Take gsoil and REW1 out of the min operator.

**Corrected**

Eq. (44) and L311. Should be Psi\_LSym , not Psi\_LApo ?

**Yes, Corrected**

L324. I suggest using a different notation for 'dt' (e.g. â t) here, to avoid the confusion with the differential operator.

Yes, thank you for this relevant suggestion. Throughout the manuscript and the appendixes, 'dt' was replaced by ' $\delta t$ ' when referring to the temporal integration

L427-428. I would use the term 'evaluation' instead of 'validation'

**Agreed, corrected**

Tab. B2. PI0 for leaf should be '-2.1'

**Corrected**

L486. Why not using an indicator of plant dessication, such as REW stem = 0.5?

This is an interesting and relevant comment. We agree with the reviewer that the water content of plant tissues is probably a better indicator of plant mortality than the percent loss of conductivity (Martinez-Vilalta *et al.* 2019, Mantova *et al.*, 2021). However, to match the abundant literature on plant hydraulic failure (*e.g.*, Adams et al. 2018), we decided that it was probably better to simulate the probability of hydraulic failure as a function of PLC in a first approximation. In addition, an accurate prediction of moisture content would require an overall integration of the carbon metabolism (Martinez-Vilalta *et al.* 2019), some processes which are currently not simulated by *SurEau-Ecos* but will be considered in future developments.

L495. Not clear how variation in gcanopy is obtained, given that three different components can be varied.

Yes, we agree with the reviewer on that point and we apologize for this unclear explanation of the setting of the sensitivity analysis. Our goal was to avoid to enter into too many details about the role of  $g_{crown}$  versus the role of  $g_s$  in the model. To clarify the results and conclusions brought by our sensitivity analyses, we performed a few changes in this section. In the new version of the manuscript, we removed the influence of  $g_{canopy}$  and only focused on  $g_{s,max}$ .

L545. Here you could add that more productive species dominate over Q. ilex in parts of the country that do not have a strong summer drought.

We thank the reviewer for this relevant comment. We added a sentence in the text to explain that while the risk of hydraulic failure was close to 0 in the temperate part of the country, where summer drought is less intense, Quercus ilex is not observed because more productive species (or cold resistant species) dominate in these areas.

**References**

- Cochard, H., Pimont, F., Ruffault, J. and Martin-StPaul, N. (202) SurEau: a mechanistic model of plant water relations under extreme drought, Ann. For. Sci., 78(2)
- Dufour-Kowalski, S., Courbaud, B., Dreyfus, P., Meredieu, C. and De Coligny, F.(2012) Capsis: An open software framework and community for forest growth modelling, Annals of Forest Science.
- Dufrêne, E., Davi, H., François, C., Le Maire, G., Le Dantec, V., & Granier, A. (2005). Modelling carbon and water cycles in a beech forest: Part I: Model description and uncertainty analysis on modelled NEE. Ecological Modelling, 185(2-4), 407-436.
- Mantova, M., Herbette, S., Cochard, H., & Torres-Ruiz, J. M. (2021). Hydraulic failure and tree mortality: from correlation to causation. Trends in Plant Science.
- Martinez-Vilalta, J., Anderegg, W. R., Sapes, G., & Sala, A. (2019). Greater focus on water pools may improve our ability to understand and anticipate drought-induced mortality in plants. *New Phytologist*, 223(1), 22-32.
- Moreaux, V., Martel, S., Bosc, A., Picart, D., Achat, D., Moisy, C., ... & Loustau, D. (2020). Energy, water and carbon exchanges in managed forest ecosystems: description, sensitivity analysis and evaluation of the INRAE GO+ model, version 3.0. Geoscientific Model Development, 13(12), 5973-6009.
- Morin, X., Bugmann, H., de Coligny, F., Martin-StPaul, N., Cailleret, M., Limousin, J. M., ... & Guillemot, J. (2021). Beyond forest succession: a gap model to study ecosystem functioning and tree community composition under climate change. Functional Ecology, 35(4), 955-975.

| Parameter                                     | Organisatio
n Level     | Importanc
e*  | Direct
availability                        | Source                                                                        | Protocol                                                        | Comments                                                                                                                           |
|-----------------------------------------------|----------------------------|------------------|-----------------------------------------------|-------------------------------------------------------------------------------|-----------------------------------------------------------------|------------------------------------------------------------------------------------------------------------------------------------|
| LAI max                            | Stand                      | High             | Yes (Remote
sensing,
inventory and      | -                                                                             | -                                                               | Dynamic parameters,
can also be related to
growth/photosynthesis                                                             |
|                                               |                            |                  | allometries)                                  |                                                                               |                                                                 | module                                                                                                                             |
| $V_L$ and $V_S$                               | Leaf and stem              | Intermediat
e | No                                            | -                                                                             | Computed from
inventories or
remote sensing               | -                                                                                                                                  |
| rfc j                              | Soil layer                 | High             | Yes (from soil                                | Hengl et al., $(2017)$                                                        | -                                                               | -                                                                                                                                  |
| $d_j$                                         |                            |                  | Partial (from soil                            | "                                                                             | -                                                               | Not available for forest                                                                                                           |
| $	heta_s$                                     | "                          | High             | No (but can
derived from soil
database) | n                                                                             | Derived from soil
texture with
pedotransfert
functions | -                                                                                                                                  |
| $\theta_r$                                    |                            | High             |                                               |                                                                               | "                                                               | -                                                                                                                                  |
| α                                             |                            | High             |                                               |                                                                               |                                                                 | -                                                                                                                                  |
| n                                             |                            | High             |                                               |                                                                               |                                                                 | _                                                                                                                                  |
| I                                             |                            | High             |                                               |                                                                               |                                                                 | _                                                                                                                                  |
| k                                             |                            | High             |                                               |                                                                               |                                                                 | _                                                                                                                                  |
| ···sat                                        | Leafand                    | Intermediat      | Yes for leaf                                  | (Bartlett et al                                                               | _                                                               | Rarely available for                                                                                                               |
| ς[, ςς                                        | stem
(symplasm)         | e                | (PV Curves)                                   | Martin-StPaul et
al., 2017;
Guillemot et al.,
2022)                  |                                                                 | stem (use leaf values
instead). Note this
parameter can be used
to inform the stomatal
conductance regulation
model |
| $\pi_{0_L}, \pi_{0_L}$                        | "                          | Intermediat
e |                                               | "                                                                             | -                                                               | "                                                                                                                                  |
| $lpha_{LApo},\ lpha_{SApo}$                   | Leaf and stem              | Intermediat
e | "                                             | "                                                                             | -                                                               | "                                                                                                                                  |
| g stom_max                         | leaf                       | Intermediat
e | Yes (gs response
curves)                   | Kattge et al.,
(2011)                                                      |                                                                 | -                                                                                                                                  |
| $\psi_{gs,50}$                                | Leaf stomata
(symplasm) | High             | II                                            | Martin-StPaul et
al., (2017);
Klein, (2014)                             |                                                                 | -                                                                                                                                  |
| $slope_{gs}$                                  | Leaf stomata
(symplasm) | Low              |                                               | II II                                                                         |                                                                 | -                                                                                                                                  |
| $g_{cuti20}$                                  | Leaf & stem
cuticle     | High             | Yes                                           | Duursma et al.,
(2019)                                                     |                                                                 |                                                                                                                                    |
| $Q_{10a}$                                     | Leaf/stem cuticle          | Intermediat
e | Partial (very few data)                       | Billon et al.,
(2020)                                                      |                                                                 |                                                                                                                                    |
| $Q_{10b}$                                     | Leaf & stem cuticle        | Low              |                                               | n í                                                                           |                                                                 |                                                                                                                                    |
| T Phase                            | Leaf & stem cuticle        | Low              |                                               |                                                                               |                                                                 |                                                                                                                                    |
| P 50                               | Leaf & stem                | High             | Yes
(Vulnerability
curve)               | Choat et al.,
2012 ; Lens et
al., 2016 ;
Martin-StPaul et
al 2017 |                                                                 | Take care of
segmentation and
methods                                                                                        |
| slope                                         | Leaf & stem                | Low              |                                               | "                                                                             |                                                                 |                                                                                                                                    |
| ,
K Plant                       | Plant                      | High             | No                                            | Mencuccini et                                                                 |                                                                 |                                                                                                                                    |
| $K_{R-SANOMON}$                               | Plant                      | -                |                                               | wi., (2017)                                                                   |                                                                 | Can be computed from                                                                                                               |
| K-SApo-LApo,max
K SApo -LApo,ma | Plant                      | -                |                                               |                                                                               |                                                                 | KPlant and hypothesis
on resistance
distribution within the
plant                                       |
| K SSym                             | Plant                      | -                | No                                            |                                                                               |                                                                 |                                                                                                                                    |
| K LSym                             | Plant                      | -                | yes                                           | Bartlett et al., (201                                                         | 6)                                                              |                                                                                                                                    |
| β                                             | Plant/Soil                 | Low              |                                               | Jackson et al., (199                                                          | 96)                                                             | At the biome scale,                                                                                                                |

---

## Author Comment (AC2)

**Reviewer 2**

The study presents a new trait-based plant hydraulics model that can scale tissue-level hydrodynamics to stand-level water use and hydraulic risks (SurEau-Ecos). The new model represents four plant water pools (leaf+stem X apoplasmic+symplasmic) and three soil water pools. The manuscript reports explorations of different numerical resolutions (explicit, implicit, semi-explicit) and recommends a time-step around 1min using implicit/semi-implicit methods. The difference between the SurEau-Ecos and SurEau, a more detailed individual-level version, is shown to be small. Sensitivity analysis suggests stand-level parameters determine the time to hydraulic failure while hydraulic traits such as psi\_50 for leaves contribute more to the drought-driven mortality risk. Finally, predictions from SurEau-Ecos at the regional scale are cross-validated with species distributions for two temperate species in France.

Overall, I really enjoy reading the manuscript partly because the equations and model structures are presented in a clear way, starting from the governing equations and then diving into different components. I appreciate the analysis of numerical schemes, which we also struggled with when developing the plant hydraulics in ED2 (and thanks for showing the biases of our semi-implicit method in a more robust way).

Meanwhile, I feel the manuscript can become more useful to the community if expanding discussions on the "necessary/optimal" complexity in plant hydraulics at ecosystem scales. In addition, lack of competition and succession can really limit the utility of the model at longer timescales in my opinion. Here are my comments following the order of the manuscript

We thank the reviewer for his positive appreciation of our work and his thorough revision of the manuscript.

We totally agree with the reviewer comment regarding the potential benefit of a discussion around the complexity of mechanisms and processes in plant hydraulic models. One of the main reasons why we developed *SurEau-Ecos* is that we aimed for a different balancing between plant representation and the possibility to apply the model for operative large-scale purpose compared to *SurEau* (Cochard *et al.*, 2021). Following the reviewer's comments and the other reviewer's general remarks, we added a new section in the manuscript that addresses to address several important points. Frist, we tried to reinforce our manuscript by providing more elements regarding the parametrization of *SurEau-Ecos*. There is now an entire section dedicated to parameter, how to determine their value and their importance in the model. Second, we took also special care to provide a more thorough evaluation of the impact of apoplasmic and symplasmic hydraulic capacitances on the model outputs. Our results showed that both the apoplasmic and symplasmic compartments had an important impact on the time to hydraulic failure and the dynamics of leaf water potentials. We added a section in the manuscript to present these new analyses and discuss the results.

We also agree with the reviewer that the lack of competition and succession can really limit the utility of the model, especially for applications that require longer time scales. The processes related to photosynthesis, respiration, growth and carbon allocation that are necessary to account for legacy effects of drought or acclimation have been overlooked there. However, we are confident that *SurEau-Ecos* could provide a comprehensive hydraulic basis for larger scale land surface, ecosystem or community models Current projects of the group aim at integrating SurEau-Ecos with the forest growth models CASTANEA (Dufrêne et al., 2005) and GO+ (Moreaux et al. 2020) and the gap model ForCEEPS (Morin et al., 2021) under the Capsis platform (Dufour-Kowalski et al., 2012). Thus, future researches and development should focus on how to link carbon and growth metabolism to hydraulic properties and how to model feedbacks between growth and hydraulic properties. We provide more details about the limitations and future developments in the new section in the manuscript.

Line 95 - Fig. 1 It is great to see energy balance of plant tissues is considered since leaf temperature can be quite a few degrees different from air temperature during drought. I was wondering whether leaf temperature dynamics have been evaluated? My experience with ED2-hydro is that it tends to overestimate leaf temperature during middays (compared with thermal camera data), which exacerbated water stress, led to more stomatal closure, less transpiration, then even higher leaf temperature.

Thank you for this remark and this suggestion. We took indeed special care to consider the energy balance of leaf tissues as it is our belief that leaf temperature is an important driver of plant stomatal and residual transpiration that should be taken in to account. Unfortunately, we did not have the opportunity so far to evaluate whether our estimations of leaf temperature were in accordance with leaf temperature measurements.

Additional comment on Fig.1. I like the idea to separate apoplasmic and symplasmic water pools, which is more realistic in terms of physiology. However, is it necessary (or in what scenarios is it necessary), and what is the additional computational cost associated with the separation? From Line 388-395, it seems the model itself is not sensitive of apoplasmic water storage. I guess the advantage is to better assimilate plant hydraulic trait measurements while I wonder what would the biases be if ignoring these water pools.

The question as to whether the separation between apoplasmic and symplasmic plant compartments affects our simulations of plant response to drought was indeed not specifically addressed in the manuscript and we thank the reviewer for raising this point. We do not agree with the reviewer on the fact that the model is not sensitive to the apoplasmic water storage as our sensitivity analyses showed the importance of *Vs* (stem water quantity) for survival time (the time between stomatal closure and plant mortality). To further evaluate how apoplasmic and symplasmic capacitances affect the general behavior of the model, we run some simulations where either apoplasmic and/or symplasmic compartments were removed (*i.e.*, set to 0) and evaluated how it affected the dynamics of plant water potentials and the time to hydraulic failure in our reference simulations. Our results (see figure below) showed that removing the apoplasmic compartments had an important impact on the time to hydraulic failure and both the dynamics of leaf water potentials. By contrast, removing the effect of the symplasmic compartments affected the infra daily temporal dynamics of leaf water potentials compared to the reference simulations but did not affect the time to hydraulic failure. This figure and the description of the results were in added the manuscript.

Line 138, I am curious about the hydraulic redistribution part. I guess it happens when psi\_soil is lower than psi\_sapo? We found that enabling water out-flow from root to soil and using the same soil-root hydraulic conductance formulation can lead to too much hydraulic redistribution that is homogenizing soil water across vertical layers. Some studies suggest that soil-root conductance can be higher than root-soil conductance (Prieto et al. 2012).

Prieto, I., C. Armas, and F. I. Pugnaire. 2012. Water release through plant roots: new insights into its consequences at the plant and ecosystem level. New Phytologist 193:830–841.

We thank the reviewer for this comment. Differences between soil-root and root-soil conductances have not been implemented in SurEau-Ecos but this is clearly some way to future improvement, in accordance with the mechanisms proposed by Prieto *et al.*, (2012).

Line 188. Why three layers? Why not making it adaptive based on total soil depth?

We chose to implement 'only' three soil layers as we considered it to be the minimum number of soil layers required to simulate plant water dynamics in complex environments, based on the results of previous water balance models (De Cáceres *al.*, 2015, Ruffault *et al.*, 2013). The first, usually rather thin, soil layer is used to compute soil evaporation. the second soil layer usually includes the soil until bedrock is reached. the third soil layer has usually an elevated rock fragment content. Adding this third soi layer may be important in water-limited environments where evidence shows that plants can expand their roots into cracks of the bedrock to get access to more water during the summer.

Line 320, Section 2.6 For numerical schemes, have you tried Runge-Kutta? In ED2, we used the fourth order RK method for integrating various PDEs, which seems to give a good balance of accuracy and computational cost.

We thank the reviewer for this suggestion. We did not try higher order schemes, as we found that the accuracy was limited by the temporal resolution of sources and sinks (in particular the fast changing of stomatal conductance with light imposes relatively small-time steps) rather than the numerical accuracy. For those time steps, we already reached the convergence with the low order scheme. However, we agree that with slower variations of these terms, a higher order scheme (as the Runge-Kutta) would surely have allowed to reach accuracy with larger time steps.

Line 520 Fig. 4. I am curious why osmotic potential plays such a minor role in all these metrics. Is it only used to convert RWC and Q? osmotic potential can have large inter- and intra- species variations (even large diurnal changes) that can change leaf turgor loss point, which is tightly associated with psi\_gs\_50. From this figure, it seems psi\_gs50 and pi\_0 are decoupled?

That is true. In *SurEau-Ecos*, stomatal closure is determined by a regulation factor ( $\gamma$ ). Several options are implemented in *SurEau-Ecos* to determine  $\gamma$ . In the version of the model presented in this manuscript,  $\gamma$  is determined according to and sigmoid function depending on the potential at 50 % of stomatal closure ( $\psi_{gs,50}$ ) and a shape parameter (*slopegs*) such as (see equation 34 in main text):

$$\gamma = 1 - \frac{1}{1 + e^{\frac{slope}{25}(\psi_{LSym} - \psi_{gs50})}}$$

This means that, with these settings,  $\pi_0$  affects the model's response to drought only through its effect on the symplasmic leaf capacitances, which play a less important role on plant water dynamics than stomatal regulation.

However, several other alternative options to determine  $\gamma$  are currently under development *in SurEau-Ecos*, including one where  $\gamma$  is a direct function of  $\pi_{TLP}$  (turgor regulation) as in Martin-StPaul *et al.*, (2017).

In addition, it is interesting to see that cuticular conductance is very important to determine survival as well. I also found the strong influence of cuticular gs on plant hydrodynamics in ED2-hydro. Are there good data sets to constrain the variations in the parameter? In general, it can be rather useful to point out which parameters can be readily acquired/measured.

We included a new section and a table in the manuscript to thoroughly describe how to parametrize the model

Line 540, typo, "the leaf and leaf ", should be "the leaf and stem"

**Corrected**

Line 595. Fig.5, the Quercus ilex result is very hard to interpret with little explanations in the text. Could it because the lack of competition in the model?

Yes, we thank the reviewer for this comment which was also raised by the other reviewer. We added a sentence in the text to explain that while the risk of hydraulic failure was close to 0 in the temperate part of the country, where summer drought is less intense, *Quercus ilex* was not observed surely because of other mechanisms that are not simulated by *Sureau-Ecos*, such as competition by more productive species, cold resistance or forest management.

Line 615-620, treating LAImax as a model parameter indicates the model only considers mature forest that has reached LAImax. This might be fine for qualitative assessment of mortality risk. However, shouldn't forests reach a new equilibrium with lower LAImax under drier conditions? (i.e. LAImax should change over time) For example, in Fig. 5, how would the mortality risk change if the forests are thinner with lower LAImax?

That is true. As discussed above, *SurEau-Ecos* do not simulate vegetation carbon fluxes and dynamics so we did not investigate such hypothesis. Couplings with forest growth models that are currently under developments will allow to explore the impact of forest dynamics on the risk of mortality, including LAI adjustments.

Tab. B1, symplasm pi\_0 for leaf should be -2.1

**Corrected**

Fig. B2-B3. Given the computational cost vary so much with longer time step, I wonder how much the difference matters at the regional scale between 1min and 10min... How worrisome we should be if models take a semi-implicit scheme with a somewhat long time-step

The reviewer is right. Depending on the applications, different time steps were implemented in the code with the assumption that, depending on the application, the balance between model accuracy and computing time may not be similar.

**References**

De Cáceres, M., Martínez-Vilalta, J., Coll, L., Llorens, P., Casals, P., Poyatos, R., ... & Brotons, L. (2015). Coupling a water balance model with forest inventory data to predict drought stress: the role of forest structural changes vs. climate changes. Agricultural and Forest Meteorology, 213, 77-90.

- Dufour-Kowalski, S., Courbaud, B., Dreyfus, P., Meredieu, C. and De Coligny, F.(2012) Capsis: An open software framework and community for forest growth modelling, Annals of Forest Science.
- Dufrêne, E., Davi, H., François, C., Le Maire, G., Le Dantec, V., & Granier, A. (2005). Modelling carbon and water cycles in a beech forest: Part I: Model description and uncertainty analysis on modelled NEE. Ecological Modelling, 185(2-4), 407-436.
- Moreaux, V., Martel, S., Bosc, A., Picart, D., Achat, D., Moisy, C., ... & Loustau, D. (2020). Energy, water and carbon exchanges in managed forest ecosystems: description, sensitivity analysis and evaluation of the INRAE GO+ model, version 3.0. Geoscientific Model Development, 13(12), 5973-6009.
- Martin-StPaul, N., Delzon, S., & Cochard, H. (2017). Plant resistance to drought depends on timely stomatal closure. Ecology letters, 20(11), 1437-1447.
- Morin, X., Bugmann, H., de Coligny, F., Martin-StPaul, N., Cailleret, M., Limousin, J. M., ... & Guillemot, J. (2021). Beyond forest succession: a gap model to study ecosystem functioning and tree community composition under climate change. Functional Ecology, 35(4), 955-975.

Ruffault, J., Martin-StPaul, N. K., Rambal, S., & Mouillot, F. (2013). Differential regional responses in drought length, intensity and timing to recent climate changes in a Mediterranean forested ecosystem. Climatic Change, 117(1), 103-